# Towards Selection Charts for Epoxy Resin, Unsaturated Polyester Resin and Their Fibre-Fabric Composites with Flame Retardants

**DOI:** 10.3390/ma14051181

**Published:** 2021-03-03

**Authors:** Noha Ramadan, Mohamed Taha, Angela Daniela La Rosa, Ahmed Elsabbagh

**Affiliations:** 1Design and Production Engineering Department, Ain Shams University, Abbassia, Cairo 11517, Egypt; noha.ramadan@eng.asu.edu.eg (N.R.); mohamed_taha@eng.asu.edu.eg (M.T.); 2Department of Manufacturing and Civil Engineering, Norwegian University of Science and Technology, 7491 Trondheim, Norway; angela.d.l.rosa@ntnu.no; 3Faculty of Engineering, Galala University, New Galala 43511, Egypt

**Keywords:** epoxy resin, unsaturated polyester resin, flame retardancy, mechanical properties, polymer matrix composites, flame retardancy index (FRI), material selection charts

## Abstract

Epoxy and unsaturated polyester resins are the most used thermosetting polymers. They are commonly used in electronics, construction, marine, automotive and aircraft industries. Moreover, reinforcing both epoxy and unsaturated polyester resins with carbon or glass fibre in a fabric form has enabled them to be used in high-performance applications. However, their organic nature as any other polymeric materials made them highly flammable materials. Enhancing the flame retardancy performance of thermosetting polymers and their composites can be improved by the addition of flame-retardant materials, but this comes at the expense of their mechanical properties. In this regard, a comprehensive review on the recent research articles that studied the flame retardancy of epoxy resin, unsaturated polyester resin and their composites were covered. Flame retardancy performance of different flame retardant/polymer systems was evaluated in terms of Flame Retardancy index (FRI) that was calculated based on the data extracted from the cone calorimeter test. Furthermore, flame retardant selection charts that relate between the flame retardancy level with mechanical properties in the aspects of tensile and flexural strength were presented. This review paper is also dedicated to providing the reader with a brief overview on the combustion mechanism of polymeric materials, their flammability behaviour and the commonly used flammability testing techniques and the mechanism of action of flame retardants.

## 1. Introduction

Polymeric materials are rapidly replacing metals and ceramic materials in various applications. This is attributed to the remarkable combination of properties like low weight, easy of fabrication and low processing temperature [1,2]. Use of polymers in the electric and electronics (E&E) industry is prominent such as in electronics housings, insulators and printed circuits [3], and similarly, in transportation industry [4,5], flexible solar cells [6] and synthetic fibres in textile industry [7,8].

Reinforcing polymers with continuous fibres like glass or carbon fibres opens a new field of applications in automotive, aerospace and construction buildings. In other words, fibre reinforcements have enabled polymeric materials to replace traditional materials like aluminium, steel and concrete that are used in high performance engineering structures [9]. High specific strength and stiffness, light weight and design flexibility are the key factors behind the continuous increase in using fibre reinforced polymer (FRP) composites [10,11]. In a commercial airplane 80–90% of the interior furnishings are manufactured from FRP [9]. Recently, FRP is used in construction and rehabilitation of metallic structures [10,12]. Constructing a FRP bridge typically reduces the weight by 75% compared to steel bridge and that is beneficial in case ground condition is poor [11].

Epoxy resin, phenolic resin, unsaturated polyester resin and vinylester resin are the most used thermosetting resins in FRP composites applications [13]. Amongst all thermosetting resins, Epoxy resins are the most widely used due to their higher mechanical properties, better adhesion to various substrates and lower shrinkage after curing compared to other resins [14]. However, longer curing time and higher cost for epoxy resin compared to polyester and vinyl ester resin hinders its use as a matrix material for automotive composites [15]. However, Epoxy composites are more appropriate for higher performance applications like aircrafts [16,17].

The ever-increasing demand for light structures and increasing fuel efficiency results in replacing more metallic parts with polymers and polymer composites. Despite the benefits of using polymeric materials, the risk of fire occurrence is increased [18,19]. The high flammability of polymers and polymer composites limits their applications and more stringent requirements should be passed for fire safety concerns [20,21]. In some studies, the reason behind the reduction in time to escape during airplane crashes, accompanied by fire, is attributed to the use of several tons of polymers in overhead bins, internal panels, seat fabric and cushions in aircraft’s passenger compartment [9,20]. Reducing the fire hazards accompanied by using polymeric materials can be achieved by incorporating flame retardants (FRs) [22]. The main applications that require flame retardants to be used in polymer composites are summarized in Table 1 [4,23,24,25].

The main functions of flame retardants are to reduce smoke and delay the time of flashover, subsequently provide sufficient time for people to escape [4]. Halogenated flame retardants were commonly used, but they were banned as they evolve toxic gases during combustion [27]. Recently halogenated FRs are replaced by phosphorus-based compounds, silicon-based compounds, borates and metal hydroxides. However, these non-halogenated flame retardants should be incorporated at high loading percentages to be effective and this in turn deteriorates the mechanical properties [16,28]. Moreover, the high loading percentages influence the resin processability. For example, the added particles increase the viscosity and the curing time for the resin and that leads to changing the processing conditions [27,29]. Thus, the challenge is to develop a flame-retardant system that enhances the fire performance of polymeric composites without deteriorating their mechanical properties.

Several review articles have analysed the different approaches that can be used to enhance the flame retardancy for polymeric materials and provide an overview of various types of flame retardant additives and their modes of action to inhibit the combustion cycle [3,23,26,30,31,32,33,34,35,36,37,38]. However, limited articles have worked on introducing quantified index to allow comparison of different flame retardant systems. Vahabi et al. [39] have proposed for a first time a universal dimensionless index known as flame retardancy index (FRI). This index helps the investigators to evaluate the performance of flame-retardant system. Vahabi et al. and Movahedifar et al. [39,40,41] have applied this index on a comprehensive set of data collected from literature to evaluate the fire performance of Polypropylene (PP), Poly (methyl methacrylate) (PMMA), Ethylene vinyl acetate (EVA), Poly (lactic acid) (PLA) and epoxy resin filled with different types of flame retardants. Moreover, literature lacks a simple selection tool that can be used to correlate between the effect of adding flame retardants on the flammability behaviour of polymeric materials and their effect on the mechanical, thermal and physical properties. Elsabbagh et al. [42] introduced a material selection chart that combines the flame retardancy performance represented by UL-94 test results with the tensile strength of natural fibre polymer composites treated with different flame retardants.

Based on the series of reliable data collected from the literature, this review concentrates on developing a variety of flame-retardant selection charts for the commonly used thermosetting polymers and their composites. These FR selection charts will relate different flame retardant test results with each other. Additionally, these charts will relate the flame retardancy performance with mechanical behaviour. These selection charts will pave a guiding tool that facilities the selection of the best FR system for different thermosetting matrix and their composites. This review is essentially divided into two sections: the first section briefly discusses the combustion mechanism, the flammability behaviour of polymers, laboratory fire testing and provides an overview of the flame retardancy mechanism and types of flame retardants. In the second section, articles on flame retardant epoxy resin and unsaturated polyester polymer matrices and their composites were reviewed and summarized in comprehensive master tables. These master tables include the type and content of flame retardant additive, cone calorimetry data including time to ignition (TTI), peak heat release rate (PHRR), total heat release (THR), calculated universal flame retardancy index (FRI) values and the available data for UL-94, limiting oxygen index (LOI) and mechanical properties represented by tensile strength and flexural strength are also included.

## 2. Combustion, Flammability and Flame Retardancy of Polymeric Materials

### 2.1. Combustion Mechanism

Understanding the combustion mechanism provides the basis of implementing an efficient flame retardant. The key contributors for the combustion process are fuel, oxygen, source of heat and chain reaction. Polymer combustion cycle starts by heating the flammable substrate till pyrolysis temperature. During pyrolysis, thermal degradation takes place and the material begins to depolymerize to unstable radicals and volatile gases. In the presence of enough oxygen and an ignition source, these volatile products act as fuel and combustion occurs. During combustion phase, toxic gases, smoke and heat are evolved. This released heat acts as a thermal feedback for further pyrolysis [13]. The combustion cycle is sustained by two main reactions, the endothermic reaction represented in thermal degradation and recycling the heat released during the exothermic reaction in the combustion phase [19,20]. Figure 1 represents a schematic diagram for the combustion process.

The structure of polymeric material, whether it is a single or double bond or aromatic structure determines the amount of energy required to break the bond and release volatile gases. The aromatic structure reduces fuel value as the chemical formula changes from C_6_ H_12_ to C_6_ H_4_ [18]. H·, O· and OH· are the most important radicals evolved from hydrocarbon flamed and these radicals participate in combustion through the following reaction H· + O_2_ → OH· + O·. This reaction generates more radicals that accelerate the burning behaviour of polymers [20].

It is worthy of note that thermosets behave differently from thermoplastics under fire. Thermoplastics undergo a reversible reaction as they melt when reheated and re-solidify when cooled. They soften when heated then flow under their own weight and drip. Dripping helps in removing heat and flame away from the bulk material. On the contrary, crosslinks in thermosets made them thermally decomposed, rather than melting [43]. Generally, thermosets are more heat-resistant compared to thermoplastics and most of them do not drip during combustion [24].

### 2.2. Flammability Behaviour of Polymers and Testing Techniques

The flammability behaviour of polymeric materials is described by several parameters such as flame spread rate, ease of ignition, time to ignition, ignition temperature, heat release rate, smoke production rate and ease of extinction [6]. There are small, medium and full-scale flammability tests applied in industrial and academic laboratories for testing manufactured products [25]. Toritzsch [24] covered the national and international fire testing regulations and procedures used for plastics and the fire regulations tests used in different applications such as building, transportation and electrical engineering. Herein, we briefly discussed the most used flammability tests on academic laboratory scale, the purpose of each test and the common test standards used for each one. These tests are underwriter laboratories test (UL-94), limiting oxygen index (LOI) and cone calorimeter. UL-94 is a rating test that measures the ignition resistance [36]. The sample is rated by V-0, V-1, V-2 or no rating after exposing it to a flame for 10 s then the flame is removed and after flame time (t_1_) is noted. The flame is applied again for another 10 s and after flame time (t_2_) is noted. The samples are rated according to the classified criteria shown in Table 2 [33].

Limiting oxygen index is the minimum concentration of oxygen in a mixture of oxygen and nitrogen that is required to maintain combustion after ignition. It is expressed in volume percentage (vol%). The oxygen percentage is 21% in air that is why materials with LOI less than 21% are considered combustible material. On the other hand, materials with LOI greater than 21% are classified as self-extinguishing. ASTM D 2863 and ISO 4589 are the standards used for this test [44].

Cone calorimeter is a bench scale test that measures the fire reaction properties. These properties are time to ignition (TTI), heat release rate (HRR), peak heat release rate (PHRR), total heat release (THR) and smoke production rate (SPR) [23]. These measured parameters are essential to assess the fire hazards of a polymeric product in a full-scale fire. ASTM E 1354 and ISO 5660 are the commonly used standards for the cone calorimeter test [44].

### 2.3. Flame Retardant Mechanisms

Combustion cycle can be divided into five stages. These stages are heating, decomposition, ignition, combustion and propagation [45]. Flame retardancy can be achieved by interrupting this cycle at any of these stages. There are three main ways to disrupt the combustion cycle. First, incorporation of additives that act as a heat sink and prevents the combustible material to reach pyrolysis temperature [30]. Second, addition of Flame Retardant (FR) compounds that produce non-flammable gases and form more char during pyrolysis. This char layer acts as an insulating layer that prevents oxygen and heat from interacting with the underlying material. The third method relies on interrupting the burning cycle during the combustion stage through adding FRs that release non-flammable gases and stable radicals that prevent progressive propagation of H· and O· free radicals and this leads to diluting the oxygen concentration in the flame zone [23,33]. Figure 2 shows the main FR mechanisms to interrupt the combustion cycle.

Generally, flame retardants can be classified based on their mechanism of action, mode of action and the functional elements that built up FR [36]. Flame retardants commonly act either in the condensed phase (phase at which the thermal degradation occurs) or in the gas phase (phase at which combustion of volatile gases occurs) [46]. In both phases, the FRs can interfere and interrupt the combustion cycle either by physical or chemical mode [19,36]. The physical mode takes place either by dilution that is achieved by reducing the concentration of decomposition gases, cooling the polymer substrate that is occurred when FR endothermically degrades and releases inert gases such as water vapor and carbon dioxide or formation of protective layer [4,23]. On the other hand, chemical mode occurs either by a dehydration process accompanied by char formation or inhibiting the high energy radicals during combustion phase [47]. FRs can also be classified either as additive when FR compounds are directly incorporated to the polymer matrix or as reactive FR when FR functional groups are part of the molecular structure of polymers [16,33,48]. In case of direct incorporation of FR, FRs are as any filler materials, the particle size and mixing conditions (whether mechanical or ultrasonication, time of mixing and temperature) are very crucial factors that affect the dispersion distribution of fillers, consequently the properties of the final system. Flame retardants are based largely on seven elements: chlorine, bromine, phosphorus, antimony, boron, nitrogen and silicon [3]. Table 3, below, summarizes different examples for the most used flame retardants and their mechanism of action [30,31,32,33,49].

## 3. Literature on Thermosetting Polymers and Their Composites Filled with FRs

The recent articles that have studied the effect of flame retardant additives on thermosets polymers and their composites have been discussed and analysed in this section. The data collected concentrate on analysing both the flammability test results and the mechanical test results for incorporating FRs into epoxy resin, unsaturated polyester resin matrix and their composites. Graphical charts are developed that can facilitate the comparison between the flammability test results obtained from literature. Moreover, flame retardant selection charts that correlate the flame retardancy performance with mechanical behaviour are also presented.

Cone calorimeter test is considered the best fire bench scale test that can simulate real state combustion of polymers. PHRR, THRR and TTI are the main results obtained from the cone calorimeter test. The hazard of fire can be evaluated by calculating the fire growth index (FGI), which is the ratio between PHRR divided by TTI and to obtain an overall fire performance of polymeric material a chart with THR (Y-axis) versus FGI (X-axis) is plotted. An increase in Y-axis value (THRR) indicates a fire of long duration. While, the increase in X-axis (FGI) value indicates a quick growth of fire [52]. The system with low THR and low FGI value gets a high safety rank. However, this plot is still a qualitative tool to evaluate the fire performance and a quantitative measure is needed. Vahabi et al. [39] have introduced a universal dimensionless index called Flame Retardancy Index (FRI), which is defined as the ratio between THR × PHRR/TTI of neat polymer and THR × PHRR/TTI of neat polymer filled with FR as shown in Equation (1).
(1)$$\mathrm{FRI}\;=\;\frac{\left[\mathrm{THR}\;\times \frac{\mathrm{PHRR}}{\mathrm{TTI}}\right]\mathrm{neat}\;\mathrm{polymer}}{\left[\mathrm{THR}\;\times \frac{\mathrm{PHRR}}{\mathrm{TTI}}\right]\;\mathrm{FR}-\mathrm{polymer}}$$

According to the value of FRI, the FR polymer system can be ranked as poor, good and excellent. From Equation (1) it can be noted that FRI with a value of one is the low limit for flame retardancy performance, below which the incorporation of FR is not effective. FRI value below one is nominated as poor, while FRI value between 10^0^ and 10^1^ is ranked as good and a system with FRI above 10^1^ is assigned as excellent.

In this section, a comprehensive data on cone calorimetry measurements (PHRR, THR, TTI), LOI, UL-94 and mechanical measurements applied on FRs, incorporated with epoxy resin, unsaturated polyester resin and their composites, were collected and summarized in master tables. From these data, the flame retardancy performance for each FR-polymer/polymer composite system was qualitatively evaluated by plotting THR versus PHRR/TTI and quantitatively ranked by calculating FRI value. These different systems were categorized as Poor, Good and Excellent based on their location in a constructed chart that combines FR weight percentage and FRI value. Moreover, the flame retardancy performance was rechecked by UL-94 and LOI test results and a graphical correlation between these different performance measures (FRI versus LOI, UL-94 versus LOI) was plotted. Additionally, flame retardant (FR) selection charts that combine the effect of FR on the flame retardancy level (UL-94 and LOI)—as well as the mechanical properties (tensile strength (TS) and flexural strength (FS)) for different FR-polymer/polymer composite systems—were constructed.

### 3.1. Epoxy Resins Containing Flame Retardants

According to the literature, the effect of various flame retardants on the flame retardancy performance as well as the mechanical properties of epoxy resin have been studied. Table 4 summarizes the data extracted from the recent research articles. Some cells are left empty since these data were not available.

The data collected in Table 4 reveal that most of the recent articles concentrated on environmentally friendly FRs and mainly focused on incorporating phosphorus based FRs into epoxy resin. Knowing that the phosphorus-based FR is effective with polymers rich in oxygen [36]. In addition, the fact that the epoxy resin is composed of glycidyl group that contains oxygen. These two facts explain the researcher’s interest to study the effect of adding phosphorus-based FR to epoxy resin. Other non-phosphorus-based FR such as mineral hydroxide and inorganic additives (Al_2_O_3_, graphene, carbon nanotubes, nano clay) are also studied. Additionally, the synergism between phosphorus and non-phosphorus-based FR is included. Figure 3 gives a brief informative view of the effect of adding FRs on the flame retardancy performance of epoxy resin. The variations of THR values versus PHRR/TTI values for different FR/epoxy resin systems are presented in Figure 3a.

Figure 3a shows the three following observations:Adding various types of FR to epoxy resin reduced both THR and PHRR/TTI ratio. As we move towards the origin of the graph, the higher flame retardancy performance for FR/epoxy resin system is obtained. Incorporating 13 wt.% synthesized nitrogen-Phosphorus-based FR DOPMP to epoxy resin shows the best fire safe system. DOPMP can act in both gaseous and condensed phases to suppress fire. In the gaseous phase during combustion DOPMP releases P· and PO· that can interact with H· and OH· free radicals. In addition, nitrogen-containing non-flammable gases are evolved and these gases dilute the concentration of flammable gases. In the condensed phase, DOPMP is decomposed into polyphosphoric acid that interacts with epoxy resin and forms a compact char [67].The variation in flame retardancy levels for neat epoxy resin is noticed in Figure 3a due to the different sources of the collected data. This variation can be explained in terms of different molecular weight and viscosity for different grades of virgin epoxy resin.DOPMP is considered, on average, the best FR system, as it is the nearest point to the origin.

Calculating the FRI value to evaluate the flame retardancy performance is more representative. Referring to Equation (1), the calculated FRI value normalized the collected data by dividing the performance of each FR/epoxy resin system by the flame retardancy performance of neat epoxy resin. FR/epoxy resin systems were categorized based on FRI and FR content in the system, as shown in Figure 3b. It was observed that poor, good and excellent performance based on the calculated FRI values for different FR/epoxy resin systems were achieved. In addition, it can be noted that most of the FR systems are located in the Good performance zone.

There are two points that have an FRI value below one and this indicates that some FR additive cannot contribute to suppressing Fire. These systems contain reduced graphene oxide (RGO) and nano-silane treated magnesium hydroxide (n-S-MgOH). In the case of reduced graphene oxide, it was noted that 1 wt.% loading has decreased the flame retardancy of neat epoxy resin. The authors [61] enhanced the flame retardancy performance by functionalizing RGO with phosphorus, nitrogen and silicon elements. However, functionalizing RGO increased the FRI value slightly above 1. Increasing the RGO content to 3 wt.% resulted in increasing FRI to reach 1.8. Yuezhan et al. [62] also reported that modifying the RGO with a covalently grafting phosphorus element in the form of a polyphosphoramide oligomer (PDMPD) and incorporating it with 1 wt.% to epoxy resin increased FRI value to 1.4. Concerning (n-S-MgOH), it was noted that 1 wt.% loading was not enough to enhance flame retardancy performance, while adding 10 wt.% increased FRI value to 1.7 [54].

It can also be observed that 13 wt.% DOPMP is in the excellent zone and that this is consistent with Figure 3a, which shows the addition of 13 wt.% DOPMP has the best fire safe rank. However, there are three more points in the excellent zone: these systems contain IFR-27-FeP-2 [53], APP_40 and APP-20_Onium ion modified nanoclay-3 [56].

Lei et al. [53] studied the flame retardancy performance of adding ferric phosphate (FeP) together with intumescent flame retardants (IFRs) composed of ammonium polyphosphate and pentaerythritol on epoxy resin. It was concluded that FeP has a synergetic effect with IFR. IFR mainly acts in the condensed phase by releasing phosphorus compounds that react with carbon source and forms a protective carbonaceous layer [19]. The addition of FeP accelerates the rate of formation of this carbonaceous layer [53] and this in return results in increasing FRI value from 6.67 for IFR with 30 wt.% loading to 11.2 for IFR incorporated with 2 wt.% FeP.

The best system that has the highest FRI value around 20 was for APP with 40 wt.% weight content. However, the high loading content of APP results in increasing resin viscosity causing difficulties during processing and deteriorates the mechanical properties. In the light of the disadvantages accompanied by adding high amount of FRs, Reija et al. [56] have investigated the effect of adding nano-clay together with ammonium polyphosphate (APP) on enhancing the flame retardancy of epoxy resin. In their study, they succeeded in enhancing the flame retardancy performance by adding just 3 wt.% of Onium ion modified nano-clay together with 20 wt.% APP. Additionally, it can be observed from Figure 3b that the mixture of nano-clay and 20 wt.% APP can reach the same FRI value of around 20 as APP with 40 wt.% loading. The enhancement of flame retardancy performance by adding nano-clay was attributed to the formation of a protective layer by the migrating of nano-clay towards the surface during combustion [33]. However, adding nano-clay alone is not sufficient to enhance flame retardancy as can be seen in Figure 3a. 3 wt.% Onium ion modified nano-clay is located to the right of pure epoxy resin, which means it goes far away from the origin. In addition, this system has reduced FRI value to 0.6.

Based on the available data, different selection charts that relate flame retardancy performance with mechanical properties of FR/epoxy resin system are provided in Figure 4. Figure 4a,b relate UL-94 and LOI results with flexural strength, respectively, for different FR/epoxy resin systems. It can be noted that, although the addition of FRs of various FR/epoxy resin systems achieved V-0, the FS for these systems was reduced. However, the addition of biobased hyperbranched polymer containing DOPO (TA-HBP) [79] with different contents ranging from 3.8 to 13.7 wt.% shows enhancement in both flame retardancy performance and mechanical properties. This enhancement in both tensile and flexural strength was attributed to the partial engagement of the-NH-group (that is present in TA-HBP structure) with epoxy matrix during curing and that results in higher cross-linking density of epoxy resin. Thus, increasing crosslinks in epoxy matrix have enhanced the mechanical properties. Figure 4c shows that the addition of TA-HBP has increased FS, but FRI value is slightly above one. In addition, it can be noted that increasing wt.% content of nitrogen/sulphur-containing DOPO based oligomer (SFG) [84] from 2% to 8% has improved both flame retardancy performance and TS. The best condition was for SFG-8 that has reached V-0, FRI value of 2.55 and 13.7% increase in TS. Additionally, Figure 4a–c show that DPPEI-30 [64] which is a reactive curing agent has enhanced the flame retardancy performance in terms of LOI, UL-94 and FRI values without a significant change in both TS and FS. Moreover, Figure 4a–c show that DOPMA-13 [67] that has high FRI value around 11.76, LOI value of 34% and achieved V-0 in UL-94 test, has reduced both FS and TS by 29.4 and 25.6%, respectively.

Referring to Figure 4c, it can be depicted that the addition of graphene oxide (GO) [75] with 0.7 wt.% reached FRI value of 3.7 and increased TS by 23.9%. However, incorporating 2 wt.% of reduced graphene oxide achieved an FRI value of only 1.24 and TS was reduced by 29.6% [82]. Decorating reduced graphene oxide with Cu_2_O and adding 2 wt.% of this decorated GO to epoxy resin increased FRI value to 2.25 and slightly reduced the TS by 5% [82]. It can also be noted that the addition of 0.5 wt.% carbon fibre treated with nitric acid (CFNA) together with 0.5 wt.% carbon nanotube (CNT) has FRI value of 2.27 and increased TS by 20.5%. However, increasing the weight content of both CFNA and CNT to 1.5 wt.% kept the value of FRI around 2.77, but it reduced TS by 16.8%.

To summarize the previous discussion, the mechanical properties for each FR/polymer system were normalized and plotted versus the flame retardant properties to provide a simple chart that can evaluate the efficiency of different FR regardless the type of epoxy resin used. Therefore, relative tensile and flexural strength were calculated based on dividing the mechanical property of FR/polymer system by the mechanical property of neat polymer. The calculated value below one means that the mechanical strength has decreased.

The charts in Figure 4d–f were divided into three regions, according to the positions of points relative to the origin. The points in the red zone refer to low flame retardancy performance and low relative mechanical strength. Meanwhile, points located in the yellow zone represent good flame retardancy performance and relative mechanical strength below one. The best FR/polymer system goes to the points located in the green zone. The points in the green zone indicate that both the flame retardancy performance and relative strength are high. The points can be easily ranked relative to each other. It is observed that TA-HBP is in the green zone throughout Figure 4d–f. Moreover, Figure 4f shows that there are three more points in the green zone. These systems are SFG-8, GO-0.7, CF-0.5 (synergized with CNT-0.5) and DPPEI-30.

### 3.2. Unsaturated Polyester Containing Flame Retardants

According to the literature, various FRs have been used with unsaturated polyester resins. Table 5 summarizes PHRR, THR and TTI, FRI, LOI, UL-94, FS and TS results of various unsaturated polyester/FR systems. 

Comparing Table 5 with Table 4, it can be noted that various additive FRs have been used and there are no reactive FRs studied with unsaturated polyester. Some of the used flame retardants are more effective with epoxy resin than unsaturated polyester resin. For example, adding intumescent flame retardant (IFR) with 32 wt.% [92] to polyester has achieved FRI of value 2.3, while adding 30 wt.% IFR [53] to epoxy resin reached an FRI value of 6.69. Additionally, hybridizing IFR with montmorillonite reduced an FRI value of 4.48 for epoxy resin, while keeping an FRI value of 2.3 in the case of polyester resin. Moreover, hybridizing 20 wt.% APP with 3 wt.% nano-clay and adding it to epoxy resin [56] has achieved FRI value of 20.2, while adding 20 wt.% APP together with 5 wt.% Na modified nano-clay to unsaturated polyester resin [90] reached an FRI value of 5. However, hybridizing 14.9 wt.% APP with 0.1 wt.% boron silicate-based graphite oxide [104] raised the FRI value of unsaturated polyester resin to 16. Another example, the addition of 15 wt.% DMPY [98] to unsaturated polyester resin, has no rate in terms of UL-94 test. In contrast, adding 12 wt.% DMPY [87] to epoxy resin achieved V-0.

Figure 5a gives a bright view for the flame retardancy performance of various FR/unsaturated polyester system. Similar to Figure 4a, the addition of FRs reduced both the THR and PHRR/TTI ratio for polyester system. It was previously mentioned that FRI value is more representative in comparing the effect of different FRs on unsaturated polyester. Figure 5b illustrates the variation in FRI values for different FRs with different weight percentages. It is obvious that almost all points are in the good zone, except four points located in the poor zone and two points located in the excellent zone. The four points in the poor zone corresponded to organic magnesium hydroxide (OMH) with 1 and 4% loading content [95] and graphite carbon nitride (g-C3N4) [96] corresponded with loading content 1 and 2%. It can be concluded that low loading of inorganic compounds has an adverse effect on the flame retardancy performance. The best FR system goes to a mixture of 0.1 wt.% of boron silicate-based graphene oxide and 14.9 wt.% of APP/MMT nano-compound [104]. This system has achieved FRI value of 16.4. It is clearly noticed that hybridizing ammonium polyphosphate (APP) with low content of nano clay or carbon-based flame retardants as graphene oxide enhanced the flame retardancy of both epoxy and unsaturated polyester resins. 

The material selection charts that correlate the flame retardancy performance with the mechanical properties for different FR/unsaturated polyester systems are shown in Figure 6 and Figure 7. The points are named according to the name of FR type and its content. These charts can be used as a simple tool to select the optimum FR that balances between flame retardancy performance and mechanical properties. Each flame retardant was evaluated according to its position relative to pure unsaturated polyester and relative to the other types of flame retardants.

From Figure 6, it is obvious that the addition of various types of FRs has a negative effect on both tensile strength and flexural strength. However, as shown in Figure 6a,c,d, the addition of APP whether by hybridizing it with 2 wt.% zinc borate and 1 wt.% montmorillonite [91], or by coating it with melamine [93], enhanced both the flame retardancy and tensile strength. Moreover, Figure 6b shows that hybridizing 20 wt.% App with 5 wt.% organic modified montmorillonite [90] increased flexural strength and achieved an FRI value of 5. Figure 6a–c can be easily used to compare the effect of adding FRs on both mechanical and flame retardancy performance in terms of FRI, UL-94 and LOI with the non-flame retarded polymer.

Based on the charts in Figure 7, the points located in the green zone can be easily ranked relative to each other and the ranking can be as follows:APP 17 wt.% (synergized with nano-clay (1% MMT)-Mineral FR (2% ZB)) [91];APP 20 wt.% (synergized with organic modified nano-clay (5% MMT) [90];APP/MMT nano compounds 14.9 wt.% (synergized with 0.1% boron silicate-based GO) [103];Melamine coated APP 10 wt.% [93].

Moreover, it can also be observed throughout Figure 7a–c that the addition of APP alone at weight content of 30% [104] achieved an FRI value of 8 and LOI value of around 35. However, this high content of APP reduced both tensile and flexural strength by approximately 50%. Additionally, it can be noted that hybridizing APP with metal oxides at a low loading content of around 0.6% increased both tensile and flexural strength by 20%, compared with adding APP alone without influencing the flame retardancy performance. Figure 7c,d further show that increasing weight content of aluminium dialkylphosphinate (Alpi) from 5 to 25% can shift the Alpi/unsaturated polyester system from the red zone to the yellow zone, but this comes at the expense of the tensile strength. The unsaturated polyester has lost 70% of its tensile strength by the addition of 25 wt.% of Alpi [98].

Comparing Figure 7c with Figure 4d it can be observed that the addition of Alpi with 12.5 wt.% [85] reduced the flexural strength of epoxy resin by only 20%. Alternatively, the addition of just 5 wt.% Alpi to unsaturated polyester resin reduced the tensile strength by 40% and increasing the Alpi content to reach 15 wt.% reduced the tensile strength by 60%. Additionally, the incorporation of Alpi had a better effect on enhancing the flame retardancy performance of epoxy resin than unsaturated polyester resin. Adding 12.5 wt.% Alpi to epoxy resin has achieved LOI value of 39.5%, while the addition of 25 wt.% Alpi to unsaturated polyester has reached 29.5% in LOI test. The higher performance achieved by the addition of Alpi to epoxy resin can be attributed to using of nano sized Alpi [85]. It is worthy to note that the preparation of FR/polymer system may explain these different results. In Refs. [85,98], the authors mixed Alpi with epoxy and unsaturated polyester resins with a mechanical mixer followed by ultrasound sonication. However, the authors of [85] sonicated the mixture for 2 h at 30 °C, while the authors of [98] sonicated the FR/polymer mixture for 40 min. Increasing the time of sonication may result in better dispersion of FR particles, and consequently, better flame retardant and mechanical properties.

FRI can be used as a reliable measure in comparing the performance of different FR/polymer system. However, this is based on the data collected from cone calorimetry, which is an expensive test compared to UL-94 and LOI test. Therefore, correlating FRI with both UL-94 and LOI test results based on the data collected from the literature can guide researchers to select which FR/polymer system condition needs to be tested using a cone calorimeter. Figure 8 and Figure 9 illustrate the flame retardancy performance in terms of FRI versus UL-94 and FRI versus LOI for FR/epoxy resin and FR/unsaturated polyester resin systems, respectively. From Figure 8a and Figure 9a, it can be depicted that, whatever the FR/polymer system, there is a direct correlation between FRI and LOI values. This finding is in agreement with previous reviews [39,40,41]. However, in the case of UL-94 results, there is no specific correlation between FRI and UL-94 results, as it can be seen in Figure 8b and Figure 9b. Moreover, it can be noted that some systems did not pass UL-94 test but achieved FRI greater than 1.

### 3.3. Fabric Composites Filled with FR Additives

Reviewing the effect of incorporating FRs into thermosetting composites is as important as reviewing the effect of FRs on the matrix only, since, for high performance applications carbon and glass fibre-fabric are commonly used as reinforcements for epoxy and unsaturated polyester resins. Moreover, as the awareness of environmental issues has increased, the number of studies investigating the possibility of replacing synthetic fibres with natural fibres in different applications have also increased. In this regard, this paper attempts to summarize and discuss the effect of adding FRs to fibre-fabric reinforced epoxy and unsaturated polyester composites.

Unfortunately, the number of research papers that studied the FR effects on thermosetting polymer composites is limited, especially for natural fibre reinforced thermosets. Table 6 gives the FR used with different fibre-fabric composites. The extracted cone calorimetry data, UL-94 and LOI results, as well as tensile and flexural strength results are also presented in Table 6. It can be noted that some composites provided in Table 6 are based on resin blends reinforced with different types of fibres. Blending of different types of resins is used as one of the techniques to reduce the flammability of polymeric materials.

To give a bright view of FR effect on flame retardancy performance of thermosetting polymer composites, the extracted data are classified according to the type of matrix, whether epoxy resin or unsaturated polyester resin and according to the type of fibre-fabric reinforcement whether glass fibre, carbon fibre or natural fibre. Figure 10 illustrates the variation of THRR with respect to PHRR/TTI for different FRs added to glass fibre-fabric epoxy composites (Figure 10a), carbon fibre-fabric epoxy composites (Figure 10c), natural fibre-fabric epoxy composites (Figure 10b) and glass fibre-fabric unsaturated polyester composites (Figure 10d). It is observed that ammonium polyphosphate (APP) is the most used FR in all types of composites. The direction of arrows in Figure 10a,b,d reveals that the addition of flame retardants enhanced the flame retardancy performance of glass and natural fibres reinforced epoxy composites as well as glass fibre reinforced unsaturated polyester composites. However, Figure 10c shows that the arrows for both Epoxy_ CF-DOPO [112] and Epoxy-CF_ Nano clay [113] systems go towards the right side (away from the origin), while the other two systems (Epoxy-FR-CF [122] and Epoxy/Cyanate ester-CF [117]) arrows’ go towards the origin. This can be attributed to the different techniques and types of FRs used in enhancing the flame retardancy. In the articles [112,113], the authors mixed the FRs with epoxy resin first, then the mixture was added to the carbon fibre. Carbon-based materials in the form of graphene grafted by DOPO(G-DOPO) [112] and carbon nanotubes [113] were used as FRs. These carbon materials the reduced time to ignition of composites. The ignition properties of polymers are affected by absorption coefficient and thermal inertia, which is the product of thermal conductivity, density and specific heat capacity [131]. It is to be noted that thermal inertia and the absorption coefficient have a contradictory effect on TTI. Time to ignition can be delayed by increasing thermal inertia and decreasing the absorption coefficient [132]. The addition of carbon materials is supposed to increase both the thermal conductivity and absorption coefficient of the composites. Regarding the ignition time, the increase in absorption coefficient induced by carbon fillers is more significant than the increase in thermal conductivity [131]. Furthermore, it can be noted that in case of adding G-DOPO to carbon/epoxy composites there is a reduction in THRR and this is due the sheet structure of G-DOPO that restrains the escape of volatile gas. Consequently, the transfer of heat to the internal matrix was hindered [112].

On the other hand, Shi et al. [122] used another technique: instead of adding FRs to the matrix, they coated the carbon fabric with a bio-based polyelectrolyte complexes (PEC) composed of chitosan (CH) and ammonium polyphosphate (APP) that acts as a flame retardant material. This PEC coat decomposed at low temperature, producing phosphorus-rich condensed char and non-flammable gases that act as a shielding layer, hindering the transfer of heat and oxygen to the unburned material. Another well-known technique that can be used to reduce the flammability of polymeric materials is the blending of different types of resins. However, only very limited research [117,119,133] has used this method with polymer composites. Martins et al. [117] studied the effect of blending epoxy resin with different ratios of cyanate ester and the addition of DOPO to these blends on the flame retardancy of carbon fabric epoxy composites. They concluded that blending has a positive effect in enhancing the flame retardancy of the composites. This is attributed to the presence of the high content of triazine structure that is responsible for the high glass transition temperature of cyanate ester (T_g_ about 400 °C) [134]. Additionally, cyanate ester has a condensed cross-linked structure, and this leads to high charring during combustion [134]. Figure 10c shows that blending 40 wt.% cyanate ester with epoxy resin and adding 9.4 wt.% of DOPO to carbon fibre composites has the lowest PHRR/TTI and THRR. Another work conducted by Kandola et al. [119] also used the blending method in enhancing the flame retardancy of unsaturated polyester reinforced with glass fabric. In their study they blended unsaturated polyester with different grades of phenolic resin. Phenolic resins are known by their high fire resistance properties. However, their brittleness limits their use in structural applications. Therefore, blending can achieve a balance between FR and mechanical properties. Figure 10d illustrates that blending 50 wt.% of phenolic resin has the best FR properties.

According to the literature, some articles have studied the effect of adding FRs of the same type and weight content on both matrix resin alone and matrix reinforced fabric composites. Thus, it was interesting to compare the obtained results by plotting THRR versus PHRR/TTI for different FR/matrix and FR/composite systems. Two points can be observed from Figure 11: First, all composite systems, whatever the kind of fabric, whether synthetic fibre-fabric or natural fibre-fabric, have lower THRR and PHRR/TTI than the matrix. Reinforcing polymeric materials with synthetic fibres such as glass and carbon fibre helps in resisting fire, as fibres displace a certain weight fraction of polymer matrix results in reducing the flammable material content. Additionally, the fibres act as an insulating layer that slows down the heat penetration to the underneath composite material [135]. Furthermore, they act as a physical barrier to volatile gases [49]. Moreover, the high thermal stability of both glass and carbon fibres made them not directly contribute to increasing the heat released [51]. Consequently, the heat released per unit volume of the composite reduces with its increasing thickness [136,137]. Comparing the fire temperature range (500–1100 °C) with the softening temperature of glass fibre (about 850 °C), it can be noted that glass fibre has high fire resistance [137]. Thus, reinforcing polymeric materials with glass fibre has a significant effect on enhancing flame retardancy.

Regarding natural fibres, their ability to char during combustion allows the formation of a charring layer that isolates the underlying polymer from the evolved heat [49]. The second point to be noted is that the addition of FRs has a more significant effect on enhancing the flame retardancy of non-reinforced polymers than composites. This behaviour was discussed by Todly et al. [138]; they suggested that the fabric reinforcements hinder the activity of the FRs of forming a well-developed charring layer.

To compare the influence of adding FRs to fibre-fabric composites, regardless of the type of matrix resin or reinforcement, FRI versus FR wt.% was plotted for different FR/composite systems, as shown in Figure 12. It can be noted that almost all points are in the good zone, except that few points are located in the poor zone and one point is located in the excellent zone. The points that are located in the poor zone correspond to graphene grafted by DOPO(G-DOPO) [112] and carbon nanotubes [113] that are added to carbon fibre reinforced epoxy resin with a low loading content of below 5%. Alternatively, the point located in the excellent zone corresponds to bromine-based FR synergized with antimony trioxide that is added to glass fibre reinforced unsaturated polyester composite [115]. 

Figure 13 and Figure 14 illustrate a number of material selection charts for different FR/composite systems. Figure 13a–d clearly shows that the addition of various FRs enhanced flame retardancy without a significant compromise in the mechanical properties. This indicates that the fibre-fabric reinforcement is the governing factor to determine the tensile and flexural strength of the composites. Figure 13d shows that flame retardancy performance (in terms of LOI) and mechanical properties (in terms of FS) for the glass fabric reinforced epoxy composite was enhanced by coating glass fabric with DOPO, instead of mixing FR with epoxy resin [128]. 

## 4. Conclusions and Future Perspectives

A survey of different FRs used with epoxy, unsaturated polyester and their composites was carried out. The effect of FRs on both flame retardancy performance and mechanical properties were considered. The universal flame retardancy index (FRI) was calculated based on the data extracted from the cone calorimetry test and this index was used to evaluate the efficiency of flame retardant performance. The main outcome of this paper is the construction of different material selection charts that combine the effect of certain FR on the flame retardancy performance in terms of FRI, LOI or UL-94, as well as mechanical properties in terms of TS, FS, relative TS and relative FS. These charts facilitate the ranking of different FRs and help in selecting the optimum systems that balance between both flame retardant and mechanical properties. Herein, the key findings of this review paper include the following:Phosphorus-based FRs proved their capability to enhance the flame retardancy of both epoxy and unsaturated polyester even at low loading below 10 wt.%. APP is the most effective phosphorus-based FR used. However, in order to achieve high FRI, APP should be loaded within the range 15–40 wt.% and this comes on the expense of mechanical properties. Therefore, synergizing APP with other additives or decorating APP with other FR compounds can reduce the required content of APP.Synergizing 20 wt.% APP with 3 wt.% of nano-clay achieved an FRI value of 20, which is the same value of loading 40 wt.% APP to epoxy resin. In addition, hybridizing 14.9% nano APP/MMT compound with 0.1 wt.% boron silicate graphene oxide achieved an FRI value of 16 for unsaturated polyester resin. Generally, the combination of different FRs is a good strategy to enhance flame retardant properties.Carbon-based fillers succeeded in acting as an FR at low loading percentage varied from 0.5 to 2 wt.%. However, they can just achieve FRI values between 1 and 2.5. The maximum attained FRI value of 3.7 was for the addition of 0.7 wt.% graphene oxide to epoxy. In addition to the flame retardant effect of carbon-based fillers, they can act as reinforcements. The addition of GO and carbon nanotubes to epoxy resin enhanced both FRI and tensile strength for epoxy resin.It can be concluded from the constructed selection charts that the mechanical properties are significantly affected by the type of FR used and its loading content. Generally, the addition of FRs reduced the mechanical properties. However, some systems enhanced both the flame retardancy performance and mechanical properties. With respect to the unsaturated polyester resin, hybridizing 20 wt.% APP with 5 wt.% nano-clay reached an FRI value of 5 and increased flexural strength by 40%. In addition, hybridizing 17 wt.% APP with 1 wt.% MMT and 2 wt.% zinc borates achieved V-0 in UL-94, an FRI value of 5 and increased tensile strength by 70%. On the other hand, increasing the APP content to 30 wt.% reduced both TS and FS by 50%. Regarding epoxy resin, the systems that enhanced both mechanical and flame retardant properties correspond to 0.7 wt.% GO (FRI value of 3.7 and relative TS value of 1.35), 8 wt.% nitrogen/sulphur-containing DOPO (FRI value of 2.5 and relative TS value of 1.2) and 7.35 wt.% biobased hyperbranched polymer-DOPO (FRI value of 1.2 and relative TS value 1.6)Reinforcing both epoxy and unsaturated polyester resins with carbon and glass fibre-fabric reduced the flammability behaviour of pristine resin. However, the addition of FRs to composite materials is not as effective as incorporating them with pure polymer. The presence of inert fabrics hinders the activity of FRs of forming a well-developed charring layer. The blending of different resins and coating of fabrics with FRs, instead of mixing them with the matrix, is a solution to enhance the flame retardant properties of the composites. The blending of epoxy resin with cyanate ester enhanced the flame retardancy performance of carbon fibre-fabric composites. Furthermore, the blending of unsaturated polyester with phenolic resin enhanced the flame retardancy performance of glass fibre-fabric composites.From the reviewed literature, it can be noted that there has not been enough research performed on studying the effect of FRs on flame retardant properties, as well as mechanical properties of epoxy resin, unsaturated polyester resin and their composites. Moreover, only very few articles have studied the effect of FRs on other properties, such as thermal conductivity, optical, sound absorption and rheological properties such as viscosity and curing behaviour. Studying the effect of FRs on different properties other than flame retardancy is necessary in order to meet the end needs of the final product. In addition, properties such as viscosity and the curing behaviour of polymers can guide the manufacturer to select the appropriate processing technique.It is recommended that future research focuses on the following points:Studying the effect of FRs on other properties, besides flame retardancy, such as mechanical, physical, optical and thermal conductivity.The idea of material selection chart should be extended to correlate properties (such as physical, optical and sound absorption), other than mechanical with flame retardant properties for different types of polymers and polymer composites. These charts will provide a quick selection tool for the production sector to select the needed FR/polymer materials that can meet the end needs of the final product.Combining data from future studies together with the data collected in this review and other reviews [39,40,41] will provide a large database and open the avenue to develop numerical models that can evaluate different aspects of flame-retarded polymers.From the environmental perspective, research should concentrate on using biobased FRs to overcome the negative impacts of FRs on human health and the environment. Moreover, use of the life cycle assessment (LCA) tool should be considered to study the impact of flame retardant polymeric products on the environment.

## Figures and Tables

**Figure 1 materials-14-01181-f001:**
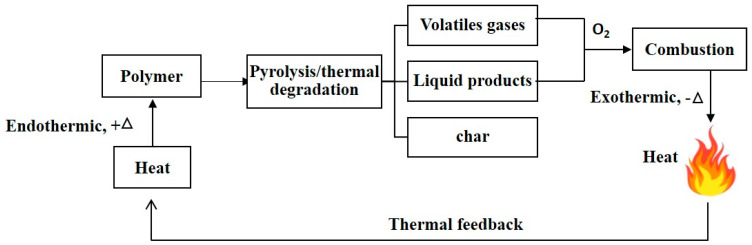
Combustion process.

**Figure 2 materials-14-01181-f002:**
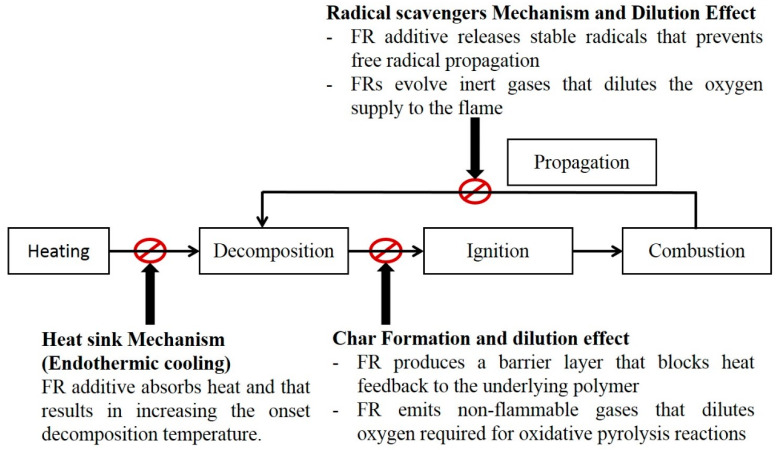
The main flame retardant mechanisms.

**Figure 3 materials-14-01181-f003:**
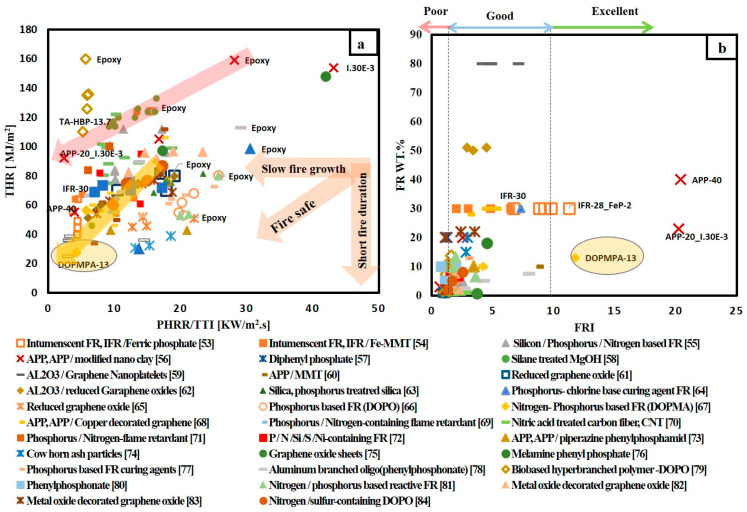
(**a**) THR versus PHRR/TTI for different FR/epoxy resin systems; (**b**) FR wt.% versus flame retardancy index (FRI) for different FR/epoxy resin systems.

**Figure 4 materials-14-01181-f004:**
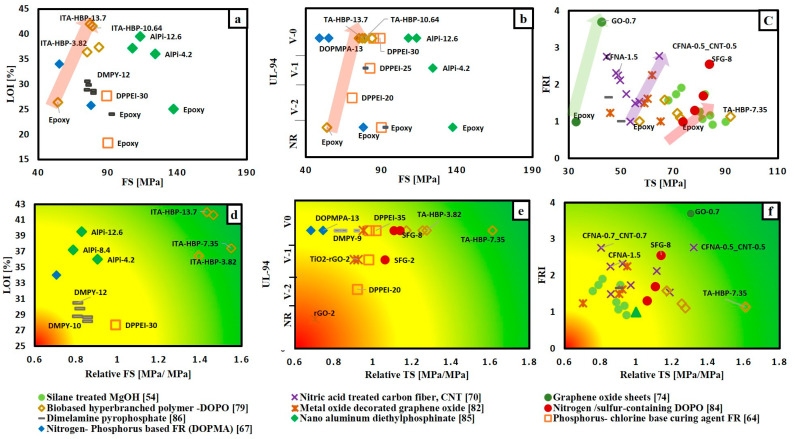
Selection chart of FR type regarding the desired flame retardance level and the mechanical property for epoxy resin. (**a**) LOI% versus FS; (**b**) UL-94 versus FS; (**c**) FRI versus TS; (**d**) LOI% versus relative FS; (**e**) UL-94 versus relative TS; (**f**) FRI versus relative TS.

**Figure 5 materials-14-01181-f005:**
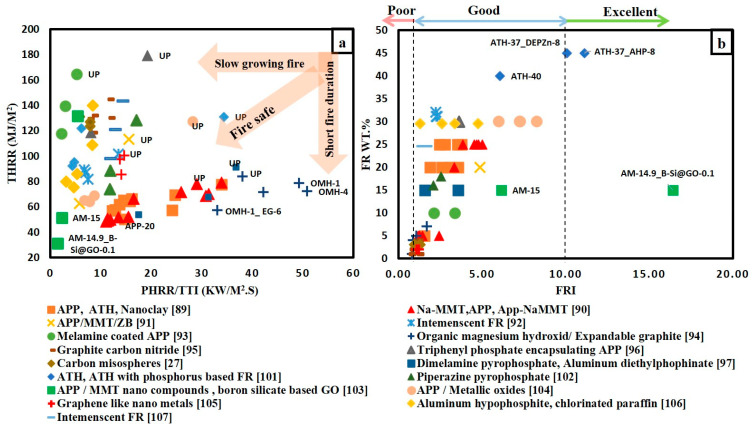
(**a**) THRR versus PHRR/TTI for various FR/Unsaturated polyester resin; (**b**) flame retardancy index (FRI) for different FR/unsaturated polyester system with respect to weight content of FRs.

**Figure 6 materials-14-01181-f006:**
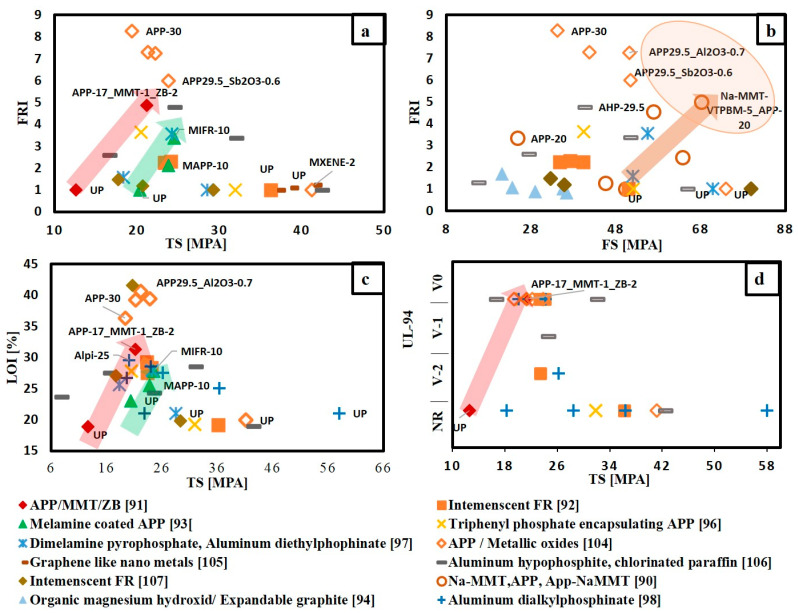
Selection chart of FR type regarding the desired flame retardance level and the mechanical property for unsaturated polyester resin. (**a**) FRI versus TS; (**b**) FRI versus FS; (**c**) LOI% versus TS; (**d**) UL-94 versus TS.

**Figure 7 materials-14-01181-f007:**
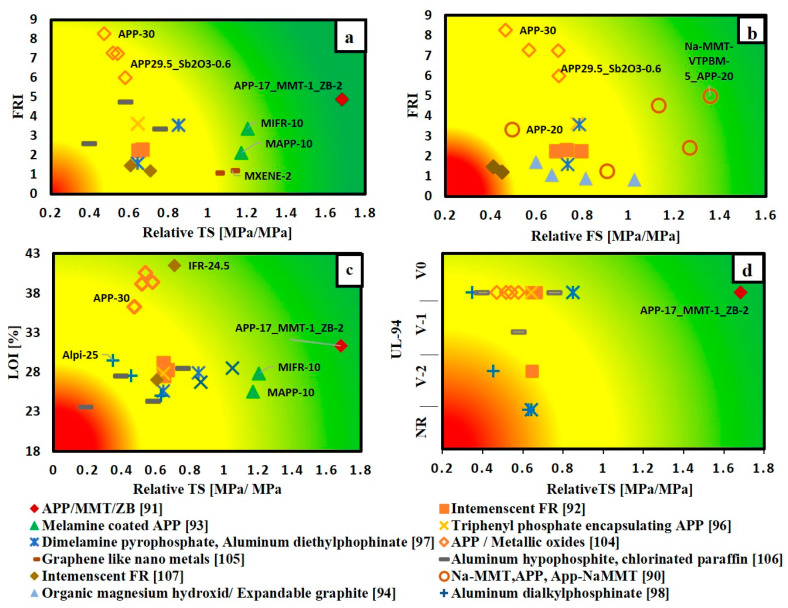
Selection chart of FR type regarding the desired flame retardance level and the relative mechanical property for unsaturated polyester resin (**a**) FRI versus relative TS; (**b**) FRI versus relative FS; (**c**) LOI% versus relative TS; (**d**) UL-94 versus relative TS.

**Figure 8 materials-14-01181-f008:**
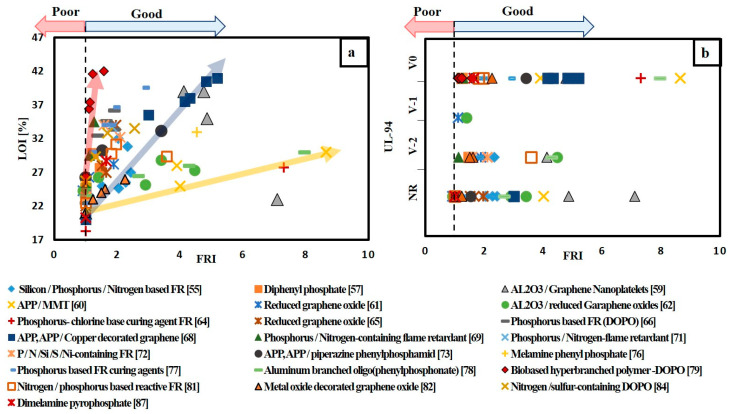
(**a**) FRI versus LOI for different FR/epoxy resin systems; (**b**) FRI versus UL-94 for different FR/epoxy resin systems.

**Figure 9 materials-14-01181-f009:**
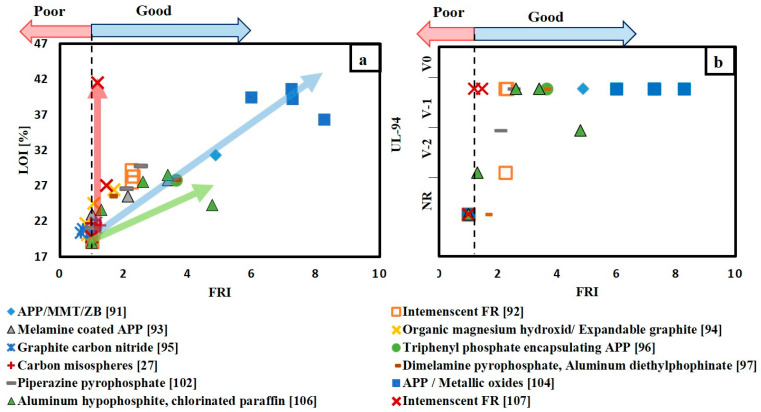
(**a**) FRI versus LOI for various FR/unsaturated polyester system; (**b**) FRI versus UL-94 for various FR/unsaturated polyester resin.

**Figure 10 materials-14-01181-f010:**
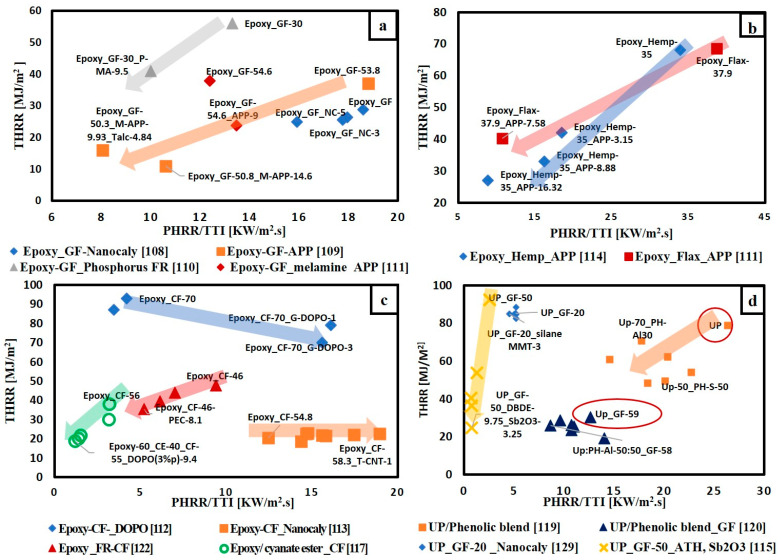
THRR versus PHRR/TTI for different composites (**a**) FR/epoxy reinforced glass fabric; (**b**) FR/epoxy reinforced natural fibre-fabrics; (**c**) FR/epoxy reinforced carbon fibre-fabrics; (**d**) FR/unsaturated polyester reinforced glass fibre-fabric.

**Figure 11 materials-14-01181-f011:**
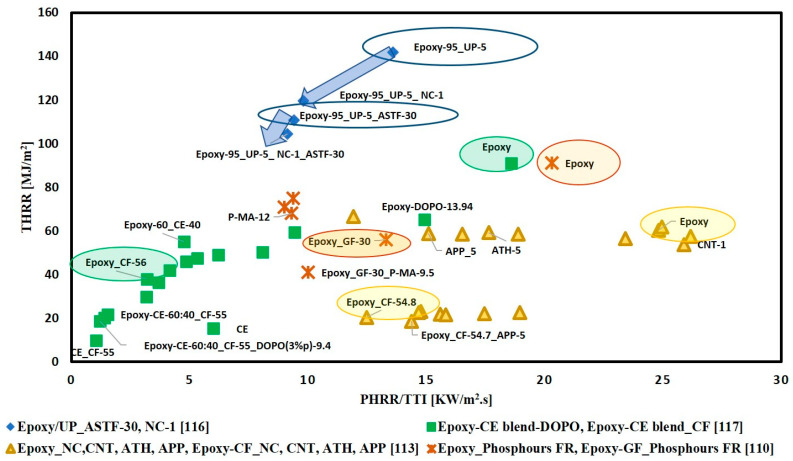
THRR versus PHRR/TTI for FR/matrix and FR/composite.

**Figure 12 materials-14-01181-f012:**
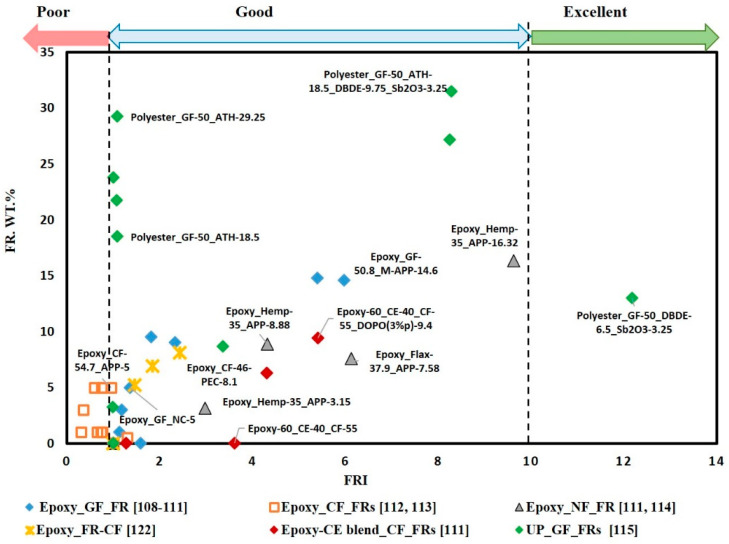
Flame retardancy index (FRI) for different FR/composite systems with respect to weight content of FR.

**Figure 13 materials-14-01181-f013:**
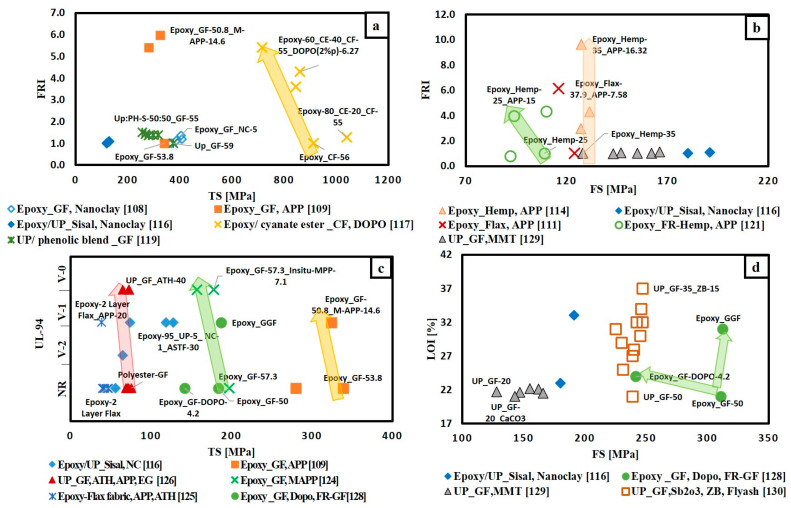
Selection charts of FR type regarding the desired flame retardance level and the mechanical property for epoxy and unsaturated polyester composites (**a**) FRI versus TS; (**b**) FRI versus FS; (**c**) UL-94 versus TS; (**d**) LOI versus FS.

**Figure 14 materials-14-01181-f014:**
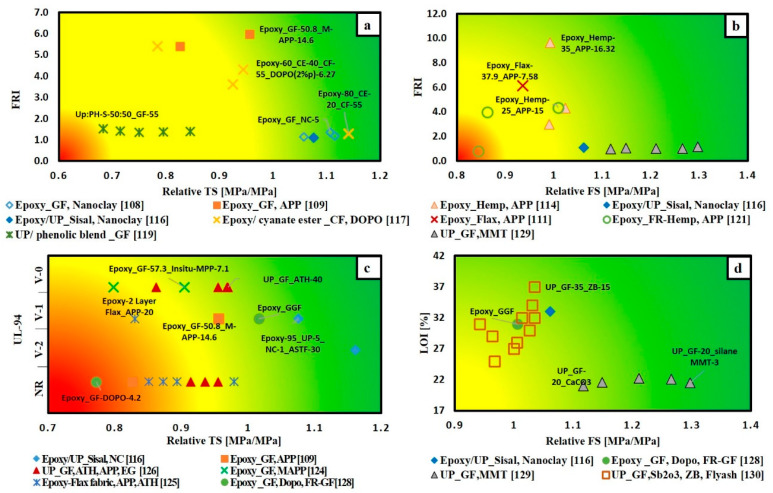
Selection charts of FR type regarding the desired flame retardance level and the mechanical property for epoxy and unsaturated polyester composites (**a**) FRI versus relative TS; (**b**) FRI versus relative FS; (**c**) UL-94 versus relative TS; (**d**) LOI versus relative FS.

**Table 1 materials-14-01181-t001:** Application of flame retardants.

Market Area	Applications	FR Governing Aspect and Standards Used
**Fabrics and apparel**	Natural fibre (cotton, wool) composites, synthetic fibre, carpets, curtain	Flame spread regulated by the limitations of ASTM D1230
**Electric and electronics**	Wire and cable, printed circuit boards, electronics housings, appliances	Ignition resistance and flame spread according to: - International Electrotechnical Commission IEC 62441, which is an open flame “candle standard” for electronics - UL 746C Guidance for individual product standards on flame rated enclosure use
**Building constructions**	Thermal insulation for roofs, facades, walls, sheetings for roofs, floor coverings, ducting and conduit, panels, linings, coverings, thermal insulating materials [foams], mattresses, furniture cushioning	Ignition resistance and containment flame spread according to: - ASTM E-84 in the United States or Single Burning Item (SBI) in the European Union (EU) [13]. - ASTM E 162 which is a small-scale test for flame spread.
**Transportation**	- Automotive (wire and cable), seats - Aircraft (panels, overhead pins), carpets, flooring - rail vehicles (compartment linings and coverings insulation, compartment interior, seats)	- Time to escape and Ignition resistance criterion according to: - Federal Motor Vehicle Safety Standard (FMVSS) No. 302 (49 CFR 571.302) that measures the flammability resistance for materials used in the interior parts of automobiles [26] - Code of Federal Regulation (CFR) 25.853 for aircraft interiors contains three types of tests, namely, vertical burning, heat release (Ohio State University calorimeter/OSU) and smoke density measurements.

**Table 2 materials-14-01181-t002:** UL-94 test criteria.

UL-94 Classification	Criteria
**V-0**	Summation of t_1_ and t_2_ < 10 s for each specimen Summation of t_1_ and t_2_ < 50 s for the five specimens No dripping
**V-1**	Summation of t_1_ and t_2_ < 30 s for each specimen Summation of t_1_ and t_2_ < 250 s for the five specimens No dripping
**V-2**	Summation of t_1_ and t_2_ < 30 s for each specimen Summation of t_1_ and t_2_ < 250 s for the five specimens Dripping allowed

**Table 3 materials-14-01181-t003:** Examples of FRs and their mechanism of action.

FR Based Element	Examples of FRs	FRs Mechanism of Action	Remarks
**Phosphorus based**	- Inorganic phosphorus FRs such as red phosphorus and ammonium polyphosphate—organic phosphorus FRs (organophosphates) include phosphate esters and phosphonate	Condensed phase - char formation enhancements Gas phase - Releasing of PO· that reacts with H· and OH·	- Not harmful, limited toxic gases evolved during combustion, can achieve good FR properties with lower loading (10–20 wt.%) compared to minerals. Relatively expensive than other FRs [50,51] - Organophosphorus compounds are the third most widely used FR. They can be used in numerous applications such as textiles, polyurethane (PU) foams, coatings and rubber [51].
**Bromine based**	Brominated bisphenols, diphenyl ethers, cyclodode ane, phenols and phthalic acids derivatives	Gas phase Releasing of bromine radical that captures the active radicals (H· and OH·)	- Low impact on polymer properties, low cost. - Release toxic gases such as dioxins and furans. These gases have a negative effect on human health and the environment. They are persistent organic pollutants (POPs) (difficult to be removed from the environment, can be easily leached out, resist degradation). - Tetrabromobisphenol A (TBBPA) is the most widely used halogenated flame retardant in printed circuit boards - Many brominated FRs have been phased out in many countries [50].
**Chlorine based**	Chlorinated paraffins and chlorinated alkyl phosphate.	Gas phase Releasing of chlorine radicals that captures the active radicals (H· and OH·)	- Toxic substances, Categorized as POPs - Water framework directive (WFD) has listed all chlorinated FRs as “priority substances” for risk assessment.
**Nitrogen based**	Melamine and melamine compounds such as melamine cyanurate, melamine polyphosphate, melamine poly (zinc/ammonium) phosphate,	Gas phase Releasing of stable nitrogen-based gases Condensed phase: Complex nitrogen compound generates cross-linked structure that promotes char formation.	- Low toxicity, low evolution of smoke. - Their efficiency lies between halogenated FR and mineral filler FR.
**Mineral fillers**	Aluminium tri-hydroxide (ATH) and magnesium hydroxides and calcium/magnesium carbonates.	Act as Heat sink	- Very cheap, non-toxic, high amounts are required to be effective (30 up to 60 wt.%) - ATH is the most used FR. It represents 40% of FR consumption
**Inorganic FRs**	Silicones, silicon oxides and transition metal oxides	Condensed phase char formation enhancements	- Very limited release of toxic gases during combustion, thermally stable

**Table 4 materials-14-01181-t004:** Cone colorimetry data (TTI, PHRR, THR), Calculated FRI value, LOI, UL-94, FS and TS for epoxy resin filled with a wide variety of FR. The designation in column two referred to FR type followed by wt.% of filler.

FR	Designation	FR wt.%	TTI (s)	PHRR (KW/m^2^)	THR (MJ/m^2^)	FRI	LOI	UL-94	FS (MPa)	TS (MPa)	Ref.
	Epoxy	0	60	923	124.2						[53]
IFR (ammonium polyphospahte)/pentaerythritol (PER) 3:1)	IFR-30	30	64	285	64.1	6.69					
IFR (ammonium polyphospahte)/pentaerythritol (PER) 3:1) and ferric phosphate (FeP)	IFR-29.5_FeP-0.5	30	46	170	56	9.23					
IFR (ammonium polyphospahte)/pentaerythritol (PER) 3:1) and ferric phosphate (FeP)	IFR-29_FeP-1	30	42	185	49.3	8.80					
IFR (ammonium polyphospahte)/pentaerythritol (PER) 3:1) and ferric phosphate (FeP)	IFR-28_FeP-2	30	39	167	39.7	11.24					
IFR (ammonium polyphospahte)/pentaerythritol (PER) 3:1) and ferric phosphate (FeP)	IFR-27_FeP-3	30	41	180	44.6	9.76					
	Epoxy	0	70	934	124.1						[54]
IFR (APP (ammonium polyphospahte)/pentaerythritol (PER) 3:1)	IFR-30	30	70	282	64.1	6.41					
IFR/organic-modified iron–montmorillonite. (Fe-OMT)	IFR-29.5_Fe-OMT-0.5	30	20	243	69	1.98					
IFR/organic-modified iron–montmorillonite. (Fe-OMT)	IFR-29_Fe-OMT-1	30	15	153	54.5	2.98					
IFR/organic-modified iron–montmorillonite. (Fe-OMT)	IFR-28_Fe-OMT-2	30	30	154	67.5	4.78					
IFR/organic-modified iron–montmorillonite. (Fe-OMT)	IFR-27_Fe-OMT-3	30	15	194	64.7	1.98					
	Epoxy	0	50	860	112		23	NR			[55]
1-oxo-4-hydroxymethyl-2,6,7-trioxa-l-phosphabicyclo[2.2.2] octane (PEPA)	PEPA-5.2	5.2	53	538	78	2.43	27	NR			
Ammonium polyphosphate (APP)	APP-2.9	2.9	61	1087	96	1.13	23.5	NR			
9,10-dihydro-9-oxa-10-phosphaphenanthrene-10-oxide (DOPO)	DOPO6.3	6.3	55	684	76	2.04	32	NR			
Octaphenyl polyhedral oligomeric silsesquioxane (OPS)	OPS-4.1	4.1	55	626	112	1.51	25	NR			
Octaphenyl polyhedral oligomeric silsesquioxane (OPS)-1-oxo-4-hydroxymethyl-2,6,7-trioxa-l-phosphabicyclo [2.2.2] octane (PEPA)	OPS-2.1_PEPA-2.6	4.7	52	524	84	2.28	25.5	NR			
Octaphenyl polyhedral oligomeric silsesquioxane (OPS)-1-oxo-4-hydroxymethyl-2,6,7-trioxa-l-phosphabicyclo [2.2.2] octane (PEPA)	OPS-2.1_PEPA-1.4	3.5	63	584	101	2.06	24.6	NR			
Octaphenyl polyhedral oligomeric silsesquioxane (OPS)-(9,10-dihydro-9-oxa-10-phosphaphenanthrene-10-oxide) (DOPO)	OPS-2.1_DOPO-3.1	5.2	55	548	83	2.33	30.8	V-1			
	Epoxy	0	43.4	1222	159						[56]
Ammonium polyphopsphate (APP)	APP-20	20	52.5	879	105	2.55					
Ammonium polyphopsphate (APP)	APP-40	40	56.4	225	55	20.40					
Onium ion modified nanoclay (Nanomer I.30E)	I.30E-3	3	29.5	1274	154	0.67					
Ammonium polyphopsphate (APP)-Onium ion modified nanoclay (Nanomer I.30E)	APP-20_I.30E-3	23	151	363	92	20.24					
	Epoxy	0	50	928	39		24.7	NR			[57]
Bisphenol A bis (diphenyl phosphate) (PBDP)	PBDP-10	10	37	567	32.3	1.46	27.6	V-1			
Bisphenol A bis (diphenyl phosphate) (PBDP)	PBDP-20	20	36	474	30.6	1.23	29.8	V-0			
	Epoxy	0	58	933	124					90	[58]
Nano silane treated Magnesium hygoxide (n-S-MgOH)	S-MgOH-1	1	55	898	133	0.92				85	
Micro size silane treated Magnesium hygoxide (m-S-MgOH)	m-S-MgOH-1	1	55	825	124	1.07				81	
Nano silane treated Magnesium hygoxide (n-S-MgOH)	n-S-MgOH-5	5	55	744	126	1.17				84	
Nano silane untreated Magnesium hygoxide (n-U-MgOH)	n-U-MgOH-5	5	56	731	120	1.27				80	
Nanosilane treate Magnesium hygoxide (n-S-MgOH)	n-S-MgOH-10	10	58	566	117	1.75				82	
Nano size untreated Magnesium hydroxide (n-U-MgOH)	n-U-MgOH-10	10	59	539	114	1.92				73	
Micro size Magnesium hygoxide (m-S-MgOH)	m-S-MgOH-10	10	58	611	120	1.58				68	
Micro size untreated Magnesium hydroxide (m-U-MgOH)	m-U-MgOH-10	10	57	572	114	1.74				71	
Al_2_O_3_	Epoxy	0	39	562	36.3		21	NR			[59]
Al_2_O_3_	Al_2_O_3_-80	80	111	326	25.1	7.10	23	NR			
Al_2_O_3_/graphene nanoplatelets (Al_2_O_3_/GNP)	Al_2_O_3_-73_GNP-7	80	113	338	36	4.86	35	NR			
Al_2_O_3_/silane graphene nanoplatelets(mGNPs)(Al_2_O_3_/SGNP)	Al_2_O_3_-73_SGNP-7	80	119	387	39	4.12	39	V1			
Al_2_O_3_/silane graphene nanoplatelets(mGNPs)/Mg(OH)_2_(Al_2_O_3_/SGNP/Mg(OH)_2_)	Al_2_O_3_-68_SGNP-7_Mg(OH)_2_-5	80	84	255	36.2	4.76	39	V0			
	Epoxy	0	50	860	112		23	NR			[60]
Ammonium Polyphosphate (APP)	APP-10	10	59	458	62	4.00	25	NR			
APP + Montmorillonite (MMT)	APP-9.4_ MMT-0.6	10	53	524	50	3.90	28	V0			
APP-Montmorillonite (MMT)	APP-MMT-10	10	60	393	34	8.65	30	V0			
	Epoxy	0	67	1138	81.6		25	NR			[61]
Reduced graphene oxide (RGO)	RGO-1	1	51	972.5	79.8	0.91	24.3	NR			
Functionalized graphene containing phosphorous, nitrogen and silicon (FRGO)	FRGO-1	1	50	891.9	69.9	1.11	26.3	V-2			
Functionalized graphene containing phosphorous, nitrogen and silicon (FRGO)	FRGO-3	3	72	753.2	70.3	1.88	28.2	V-1			
	Epoxy	0	67	1138	81.6		25	NR			[62]
Reduced graphene oxide (RGO)	RGO-1	1	51	972.5	79.8	0.91	24.3	NR			
Functionalized reduced graphene oxide polyphosphoramide oligomer (PFR-fRGO)	PFR-fRGO-1	1	64	853.3	74.4	1.40	26.3	V-2			
Al_2_O_3_	Al_2_O_3_-50	50	111	802.7	56.3	3.40	28.8	NR			
Al_2_O_3_/reduced graphene oxide (RGO)	Al_2_O_3_-50_RGO-1	51	97	775	60	2.89	25.2	NR			
Al_2_O_3_/functionalized reduced graphene oxide polyphosphoramide oligomer (PFR-fRGO)	Al_2_O_3_-50_PFR-fRGO-1	51	88	533.5	51.2	4.46	27.3	V-1			
	Epoxy	0	59.6	1397	81.3						[63]
Mesoporous silica (SH-mSiO2)	SH-mSiO2-2	2	62.4	1117	77.8	1.37					
Hyperbranched polyphosphate acrylate (HPPA).	HPPA-2	2	59	1097	75.4	1.36					
Mesoporous silica with Hyperbranched polyphosphate acrylate HPPA-SH-mSiO_2_	HPPA-SH-mSiO_2_-2	2	62.4	995.3	68.3	1.75					
	Epoxy	0	58	1770	98.5		18.3	NR	89.8	52.6	[64]
DPPEI curing agent synthesized via reaction between diphenylphosphinic chloride (DPPC) and polyethylenimine (PEI)	DPPEI-30	30	47	645	30	7.30	27.7	V-0	88.9	51.5	
	Epoxy	0	50	1103	50.91		22	NR			[65]
Polyaniline PANI	PANI-3	3	56	834	45.6	1.65	27	NR			
Reduced graphene--polyaniline (RGO-PANI)	RGO-PANI-3	3	59	845	51.82	1.51	28	NR			
Graphene-polyaniline/nickel hydroxide (RGO-PANI/Ni(OH)2)	RGO-PANI/Ni(OH)2-3	3	52	661	45.15	1.96	34	NR			
	Epoxy	0	47	1208	80.3		22.5	NR			[66]
Phosphorus–nitrogen-containing FR synthesized via reaction between 9,10-dihydro-9-oxa-10-phosphaphenanthrene-10-oxide (DOPO) and cyanuric chloride.	DOPO-2.34	2.34	38	836	68.2	1.38	32.5	NR			
Phosphorus–nitrogen-containing FR synthesized via reaction between 9,10-dihydro-9-oxa-10-phosphaphenanthrene-10-oxide (DOPO) and cyanuric chloride.	DOPO-4.67	4.67	36	727	61.8	1.65	34.6	V-1			
Phosphorus–nitrogen-containing FR synthesized via reaction between 9,10-dihydro-9-oxa-10-phosphaphenanthrene-10-oxide (DOPO) and cyanuric chloride.	DOPO-6.99	6.99	32	629	55.13	1.90	36.2	V-1			
Phosphorus–nitrogen-containing FR synthesized via reaction between 9,10-dihydro-9-oxa-10-phosphaphenanthrene-10-oxide (DOPO) and cyanuric chloride.	DOPO-9.34	9.34	30	613	53.2	1.90	33.4	V-0			
	Epoxy	0	59	1063	76.1		25.8	NR	78	82	[67]
Piperazine phosphaphenanthrene (DOPMPA)	DOPMPA-10	10	68	393	56.3	4.21	29	NR			
Piperazine phosphaphenanthrene (DOPMPA)	DOPMPA-13	13	67	285	27.4	11.76	34	V-0	55	61	
		0	62	1075	106		20	NR			[68]
Ammonium polyphosphate (APP)	APP-28	28	62	558	68	3.00	35.5	NR			
Ammonium polyphosphate (APP)/graphene nanosheets (GNS)	APP-28_GNS-2	30	59	567	46	4.16	37.5	V-0			
Ammonium polyphosphate (APP)/copper decorated graphene oxide (Cu^2+^-GO)	APP-28_Cu^2+^-2	30	56	355	60	4.83	40.5	V-0			
Ammonium polyphosphate (APP)/copper decorated reduced graphene oxide (Cu^2+^-RGO)	APP-28_Cu^2+^-RGO-2	30	55	418	56	4.32	38	V-0			
APP/copper oxide modified graphene nanotubes (CuO-GNS)	APP-28_CuO-GNS-2	30	59	380	55	5.19	41	V-0			
	Epoxy	0	90	893.8	64.1		25.5	NR			[69]
Hyperbranched phosphorus/nitrogen-containing flame retardant (HPNFR)	HPNFR-2	2	88	817.9	61.1	1.12	29.5	V-1			
Hyperbranched phosphorus/nitrogen-containing flame retardant (HPNFR)	HPNFR-4	4	82	743.9	55.2	1.27	34.5	V-0			
	Epoxy	0	52	971.7	98.8				69.3	53.5	[70]
oxidation treated CF using concentrated nitric acid (CFNA)	CFNA-0.5	0.5	69	792.7	92.5	1.74			89.3	51.9	
oxidation treated CF using concentrated nitric acid (CFNA)	CFNA-0.7	0.7	80	722.6	88.2	2.32			65.08	48	
oxidation treated CF using concentrated nitric acid (CFNA)	CFNA-1	1	62	840.2	88.9	1.53			61.84	57	
oxidation treated CF using concentrated nitric acid (CFNA)	CFNA-1.5	1.5	98	793.3	101.7	2.24			57.7	48.8	
Carbon fiber treated with concentrated nitric acid/Carbon nanotube (CNT)(CFNA-CNT)	CFNA-0.5_CNT-0.5	1	73	648.1	75	2.77			80.1	64.5	
Carbon fiber treated with concentrated nitric acid/Carbon nanotube (CNT)(CFNA-CNT)	CFNA-0.5_CNT-1	1.5	92	937	122	1.49			62.37	55.38	
Carbon fiber treated with concentrated nitric acid/Carbon nanotube (CNT)(CFNA-CNT)	CFNA-0.7_CNT-0.7	1.4	76	635	80.3	2.75			103.7	44.5	
Carbon fiber treated with concentrated nitric acid/Carbon nanotube(CNT)(CFNA-CNT)	CFNA-1_CNT-0.5	1.5	80	701.7	99.3	2.12			76.6	49.6	
	Epoxy	0	71	654.3	100.3		25.7	NR			[71]
Phosphaphenanthrene group and tetrazole ring, 6-(((1H-tetrazol-5-yl) amino)(4-hydroxyphenyl)methyl)dibenzo[c,e][1,2]oxaphosphinine 6-oxide (ATZ)	ATZ-6	6	81	482.5	83.9	1.85	33.7	V-0			
	EP	0	72	1010	95		23.3	NR			[72]
N-substituted bis(diphenylphosphanyl) amine RN(PPh2)2 (PNP)	PNP-7	7	54	748	61	1.58	34	V-1			
mononuclear nickel(II) ethanedithiolate complexe RN(PPh2)2Ni(SCH2CH2S) (PNS)	PNS-7	7	67	520	82	2.09	32.2	V-1			
	EP	0	59	1063	76.1		26.2	NR			[73]
Ammonium polyphosphate (APP)	APP-10	10	36	754	42.8	1.53	30.2	NR			
Ammonium polyphosphate (APP)/piperazine phenylphosphamide) (BPOPA)	APP-7.5_BPOPA-2.5	10	61	576	42.6	3.41	33.1	V-0			
	Epoxy	0	63.9	556.5	308.6						[74]
Cow horn ash particles (CHAp)	CHAp-5	5	81.8	455.5	301	1.60					
Cow horn ash particles (CHAp)	CHAp-10	10	80.9	405.4	289.9	1.85					
Cow horn ash particles (CHAp)	CHAp-15	15	85.9	397.1	206.9	2.81					
Cow horn ash particles (CHAp)	CHAp-20	20	86.9	392.7	203.5	2.92					
	Epoxy	0	40	1678	148					32.6	[75]
Graphene oxide sheets (GO)	GO-0.7	0.7	49	844.7	97.3	3.70				42.5	
	Epoxy	0	60	1073	76		25.6	NR			[76]
Melamine phenyl phosphate (MAPPO)	MAPPO-18	18	68	443	46	4.54	33	V-0			
	Epoxy	0	47	1208	80.6		22.5	NR			[77]
10-dihydro-9-oxa-10-phosphaphenanthrene-10-oxide (DOPO))	DOPO-7	7	41	833	66.7	1.53	34.1	V-1			
Phosphaphenanthrene/benzimidazole containing flame retardant curing agent (DTA-0.25 P)	DTA-3.2	3.2	43	1063	72.5	1.16	30.1	NR			
Phosphaphenanthrene/benzimidazole containing flame retardant curing agent (DTA-0.5 P)	DTA-6.4	6.4	42	766	64.2	1.77	34.1	V-1			
Phosphaphenanthrene/benzimidazole containing flame retardant curing agent (DTA-0.75 P)	DTA-9.6	9.6	40	712	61.1	1.90	36.7	V-0			
Phosphaphenanthrene/benzimidazole containing flame retardant curing agent (DTA-1.0 P)	DTA-12.8	12.8	38	524	52.9	2.84	39.6	V-0			
	Epoxy	0	49	1425	112.9		23.5	NR			[78]
Aluminum branched oligo(phenylphosphonate) (AHPP)	AHPP-2.5	2.5	66	907	89.6	2.67	26.5	NR			
Aluminum branched oligo(phenylphosphonate) (AHPP)	AHPP-5	5	67	744	69.1	4.28	28	V-1			
Aluminum branched oligo(phenylphosphonate) (AHPP)	AHPP-7.5	7.5	73	454	66.4	7.95	30	V-0			
	Epoxy	0	120	678.7	159.9		26.4	NR	54	57	[79]
Itaconic anhydride hyperbranched polymer (ITA-HBP)	TA-HBP-3.82	3.82	102	618.6	135.7	1.10	36.4	V-0	75.2	72.6	
Itaconic anhydride hyperbranched polymer (ITA-HBP)	TA-HBP-7.35	7.35	96	564.5	135.3	1.14	37.4	V-0	83.7	91.8	
Itaconic anhydride hyperbranched polymer (ITA-HBP)	TA-HBP-10.64	10.64	91	534	125.9	1.22	41.6	V-0	79	71.5	
Itaconic anhydride hyperbranched polymer (ITA-HBP)	TA-HBP-13.7	13.7	90	468	110.2	1.58	42	V-0	77.3	66.7	
	Epoxy	0	58	714	76.3						[80]
Phenylphosphonate (EHPP)	EHPP-5	5	44	548	70.4	1.07					
Phenylphosphonate (EHPP)	EHPP-10	10	34	584	72	0.76					
Ionic complexation between phytic acid and a novel phenylphosphonate (EHPP@PA5)	EHPP-PA5-5	5	36	294	73.7	1.56					
Ionic complexation between phytic acid and a novel phenylphosphonate (EHPP@PA5)	EHPP-PA6-10	10	37	258	69.1	1.95					
	Epoxy	0	47	1208	80.2		22.5	NR			[81]
Aminobenzothiazole-substituted cyclotriphosphazene (ABCP-0.6 P)	ABCP-0.6 P-6.6	6.6	46	465	57	3.58	29.4	V-1			
Aminobenzothiazole-substituted cyclotriphosphazene (ABCP-0.9 P)	ABCP-0.9 P-9.9	9.9	29	616	53.7	1.81	29.8	V-0			
Aminobenzothiazole-substituted cyclotriphosphazene (ABCP-1.2 P)	ABCP-1.2 P-13.3	13.3	28	559	52.7	1.96	31.2	V-0			
	Epoxy	0	55	1286	96.4		21	NR		65.1	[82]
Reduced graphene oxide (rGO)	rGO-2	2	45	849	96.7	1.24	23	NR		45.8	
Metal-based nanoparticles decorated reduced graphene oxide (TiO_2_-rGO)	TiO_2_-rGO-2	2	60	875	95.7	1.62	24.5	V-1		60.1	
Metal-based nanoparticles decorated reduced graphene oxide (Cu_2_O-rGO)	Cu_2_O-rGO-2	2	53	643	82.5	2.25	26	V-0		61.9	
metal-based nanoparticles decorated reduced grapheneoxide (Ag-rGO)	Ag-rGO-2	2	49	804	91.7	1.50	24	V-1		58.9	
	Epoxy	0	67	1138	81.6			NR			[83]
Reduced graphene oxide (RGO)	RGO-2	2	57	913.7	77	1.12		NR			
Reduced graphene oxide coated with Ni (OH)2 (RGO – Ni (OH)2-2)	RGO-Ni(OH)2-2	2	55	777.7	74.6	1.31		V-2			
Hexagonal boron nitride (hBN)	hBN-20	20	45	845.3	68.7	1.07		NR			
Hexagonal boron nitride (hBN)/Reduced graphene oxide (RGO)	hBN20_-RGO-2	22	80	743.7	63	2.37		V-2			
Hexagonal boron nitride (hBN)/Reduced graphene oxide coated with Ni (OH)2 (RGO-Ni(OH)2-2)	hBN-20_RGO coated Ni(OH)2-2	22	103	756.8	54	3.49		V-1			
	Epoxy	0	55	949	86.8		25.7	NR		73.6	[84]
Nitrogen/sulfur-containing DOPO based oligomer (SFG)	SFG-2	2	57	851	76.8	1.31	29.3	V-1		78.1	
Nitrogen/sulfur-containing DOPO based oligomer (SFG)	SFG-5	5	60	706	74.9	1.70	32.8	V-0		81.4	
Nitrogen/sulfur-containing DOPO based oligomer (SFG)	SFG-8	8	65	634	60.2	2.55	33.5	V-0		83.7	
	Epoxy	0					25	NR	137.2		[85]
Nano aluminum diethylphosphinate (AlPi) (Phousphours content-1 %)	AlPi-4.2	4.2					36	V-1	124		
Nano aluminum diethylphosphinate (AlPi) (Phousphours content–2 %)	AlPi-8.4	8.4					37.2	V-0	107.8		
Nano aluminum diethylphosphinate (AlPi) Phousphours content–3 %)	AlPi-12.6	12.6					39.5	V-0	113.3		
	Epoxy	0	58	670	123.2		20.3	NR	91.1	50	[86]
Dimelamine pyrophosphate (DMPY)	DMPY-8	8					28.2	V-1	77.9	45.1	
Dimelamine pyrophosphate (DMPY)	DMPY-9	9	40	458	75.4	1.65	28.7	V-0	77.8	45.27	
Dimelamine pyrophosphate (DMPY)	DMPY-10	10					28.8	V-0	73.3	41.23	
Dimelamine pyrophosphate (DMPY)	DMPY-11	11					29.8	V-0	74.45	42	
Dimelamine pyrophosphate (DMPY)	DMPY-12	12					30.5	V-0	73.6	40.1	
	Epoxy	0					19		74.5	40.6	[87]
Modified montmorillonite (MMT) clay	MMT-1	1					21.3		76.2	50.3	
Modified montmorillonite (MMT) clay	MMT-2	2					25.4		78.5	53.1	
Modified montmorillonite (MMT) clay	MMT-3	3					24.3		81.3	53.8	
Modified montmorillonite (MMT) clay	MMT-4	4					27.2		75.8	48.2	
Diglycidylphenylphosphate (DGPP)/Modified montmorillonite (MMT) clay	DGPP-5_MMT-1	6					29.6		88.2	55.4	
Diglycidylphenylphosphate (DGPP)/Modified montmorillonite (MMT) clay	DGPP-5_MMT-2	7					31.2		95.4	56.2	
Diglycidylphenylphosphate (DGPP)/Modified montmorillonite (MMT) clay	DGPP-5_MMT-3	8					30.6		97.6	57.1	
Diglycidylphenylphosphate (DGPP)/Modified montmorillonite (MMT) clay	DGPP-5_MMT-4	9					32		90.4	56.4	
Diglycidylphenylphosphate (DGPP)/Modified montmorillonite (MMT) clay	DGPP-10_MMT-1	6					32.1		102	62.4	
Diglycidylphenylphosphate (DGPP)/Modified montmorillonite (MMT) clay	DGPP-10_MMT-2	7					33.2		104	64.8	
Diglycidylphenylphosphate (DGPP)/Modified montmorillonite (MMT) clay	DGPP-10_MMT-3	8					33.6		109	62.8	
Diglycidylphenylphosphate (DGPP)/Modified montmorillonite (MMT) clay	DGPP-10_MMT-4	9					33.9		94.2	59.1	
Diglycidylphenylphosphate (DGPP)/Modified montmorillonite (MMT) clay	DGPP-15_MMT-1	6					31.2		78.9	52.4	
Diglycidylphenylphosphate (DGPP)/Modified montmorillonite (MMT) clay	DGPP-15_MMT-2	7					33.3		83.2	47.1	
Diglycidylphenylphosphate (DGPP)/Modified montmorillonite (MMT) clay	DGPP-15_MMT-3	8					35.1		75.6	42.3	
Diglycidylphenylphosphate (DGPP)/Modified montmorillonite (MMT) clay	DGPP-15_MMT-4	9					34.5		69.2	40.1	
	Epoxy	0					22.4	NR			[88]
Polyaniline (PANI)	PANI-1	1					22.6	NR			
Polyaniline (PANI)	PANI-2	2					24.7	NR			
Polyaniline (PANI)	PANI-3	3					25.2	NR			
Polyaniline (PANI)	PANI-4	4					25.6	NR			
Polyaniline (PANI)	PANI-5	5					25.7	NR			
Phosphorus-containing polyaniline (*p*-PANI)	*p*-PANI-1	1					23.1	NR			
Phosphorus-containing polyaniline (*p*-PANI)	*p*-PANI-2	2					25.5	NR			
Phosphorus-containing polyaniline (*p*-PANI)	*p*-PANI-3	3					29.3	V-2			
Phosphorus-containing polyaniline (*p*-PANI)	*p*-PANI-4	4					30.8	V-0			
Phosphorus-containing polyaniline (*p*-PANI)	*p*-PANI-5	5					31.1	V-0			

**Table 5 materials-14-01181-t005:** Cone colorimetry data (TTI, PHRR, THR), calculated FRI value, LOI, UL-94, FS and TS for unsaturated polyester filled with a wide variety of FR. The designation in the second column refers to FR type followed by wt.% of filler.

FR	Designation	FR wt.%	TTI (s)	PHRR (KW/m^2^)	THR (MJ/m^2^)	FRI	LOI	UL-94	FS (MPa)	TS (MPa)	Ref.
	Unsaturated polyester	0	34	1153	77.5						[89]
Na-Nano clay-25A	Na-Nclay-5	5	36	887	69.3	1.54					
Ammonium polyphosphate (APP)	APP-20	20	31	456	50.1	3.57					
Melamine phosphate (NH)	NH-20	20	33	451	61.8	3.11					
Dipentaerythritol/melamine phosphate intumescent mixture (NW)	NW-20	20	30	722	57.4	1.90					
Alumina trihydrate (ATH)	ATH-20	20	38	597	64.5	2.59					
Na-Nano clay-Ammonium polyphosphate (APP)	Na-Nclay-5_APP-20	25	36.9	453	56.9	3.76					
Na-Nano clay-Melamine phosphate (NH)	Na-Nclay-5_NH-20	25	40.1	580	65.3	2.78					
Na-Nano clay-Dipentaerythritol/melamine phosphate intumescent mixture (NW)	Na-Nclay-5_NW-20	25	41.5	670	66.4	2.45					
Na-Nano clay-Alumina trihydrate (ATH)	Na-Nclay-5_ATH-20	25	40.1	515	57.9	3.53					
	unsaturated polyester	0	34	1153	79				50.3		[90]
Na-MMT (Montmorillonite clay)-Vinyl triphenyl phosphonium bromide modifier	Na-MMT-VTPBM-5	5	45	743	66.5	2.44			63.7		
Na-MMT (Montmorillonite clay)-Vinyl benzyl trimethyl ammonium chloride modifier	Na-MMT-VTACM-5	5	34	1045	68.8	1.27			45.5		
Na-MMT (Montmorillonite clay) Hexadecyl trimethyl ammonium chloride	Na-MMT-HDTACM-5	5	32	1002	70	1.22			42.1		
Na-MMT (Montmorillonite clay) Dodecyl ethyl dimethyl ammonium bromide	Na-MMT-DWDABM-5	5	40	1034	71.7	1.45			38.2		
Na-MMT-N,N-dimethyl-N,N-dioctadecyl quaternary ammonium bromide	Na-MMT-NDQAB-5	5	33	958	77.9	1.18					
Ammonium polyphosphate (APP)	APP-20	20	31	478	52.2	3.33			24.7		
Na-MMT (Montmorillonite clay)-Vinyl triphenyl phosphonium bromide modifier/APP	Na-MMT-VTPBM-5_APP-20	25	38	419	48.5	5.01			68.2		
Na-MMT (Montmorillonite clay)-Vinyl benzyl trimethyl ammonium chloride modifier/APP	Na-MMT-VTACM-5_APP-20	25	36	426	49.9	4.54			56.9		
Na-MMT (Montmorillonite clay) Hexadecyl trimethyl ammonium chloride/APP	Na-MMT-HDTACM-5_APP-20	25	38	434	49.1	4.78			49.3		
Na-MMT (Montmorillonite clay) Dodecyl ethyl dimethyl ammonium bromide/APP	Na-MMT-DWDABM-5_APP-20	25	36	484	51.6	3.86			45.1		
Na-MMT-N,N-dimethyl-N,N-dioctadecyl quaternary ammonium bromide/APP	Na-MMT-NDQAB-5_APP-20	25	34	384	50.6	4.69					
	Unsaturated polyester	0	37.3	581	113.1		18.9	NR	50.9	12.6	[91]
Ammonium polyphosphate (APP)-Montmorillonite (MMT)-Zinc borate (ZB)	APP-17_MMT-1_ZB-2	20	48.5	280	62.7	4.867	31.3	V-0	44.2	21.2	
unsaturated polyester	unsaturated polyester	0	45	605.7	101.6		19.1	NR	50.9	36.2	[92]
IFR Intumescent Flame retardant (APP/pentaerythritol (PER)/melamine (Mel))(6:1:1)	IFR-32	32	37	259.2	87.1	2.24	27.5	V-2	34.7	23.4	
IFR Intumescent Flame retardant (APP/pentaerythritol (PER)/melamine (Mel)) (6:1:1)/MMT	IFR-29.6_MMT-1.5	31.1	32	213.1	89.2	2.30	28.3	V-0	37.1	24.1	
IFR Intumescent Flame retardant (APP/pentaerythritol (PER)/melamine (Mel)) (6:1:1)/PA-MMT	IFR-29.6_PA-MMT-1.6	31.1	30	222.9	81.8	2.25	29.2	V-0	40.3	23.3	
	unsaturated polyester	0	120	623.7	164.5		23			20.37	[93]
Melamine resin-coated ammonium polyphosphate (MAPP)	MAPP-10	10	160	461.9	139.5	2.12	25.5			23.83	
Tannic acid–iron A mussel-inspired intumescent flame retardant (MIFR)	MIFR-10	10	109	234.4	117.7	3.38	27.8			24.54	
	unsaturated polyester	0	22.6	860	84		20.5		35.5		[94]
Organic magnesium hydroxide (oMH)	OMH-1	1	17.1	842	78.9	0.82	21.7		36.4		
Organic magnesium hydroxide (oMH)	OMH-4	4	15.8	803	72.2	0.87	20.1		28.9		
Organic magnesium hydroxide (oMH)/Expandable er graphene (EG)	OMH-1_ EG-4	5	8.8	371	71.7	1.06	24.5		23.6		
Organic magnesium hydroxide (oMH)/Expandable er graphene (EG)	OMH-1_ EG-6	7	8.5	281	57.4	1.68	26.4		21.1		
	unsaturated polyester resin	0	62	520.1	131.9		19.8				[95]
Graphite carbon nitride g-C3N4	g-C3N4-1	1	53	607.5	144.6	0.67	20.4				
Graphite carbon nitride g-C3N4	g-C3N4-2	2	42	490.5	130.1	0.73	20.9				
Metal-organic framework MIL-53 (Fe)@ C on surface of graphite carbon nitride (MFeCN)	MFeCN-1	1	51	383.3	129.3	1.14	21				
Metal-organic framework MIL-53 (Fe)@ C on surface of graphite carbon nitride (MFeCN)	MFeCN-4	4	38	313	118.4	1.13	21.8				
	unsaturated polyester	0	39	750.6	179.3		19.2	NR	52.1	31.9	[96]
Diatomite/ammonium polyphosphate encapsulated in Triphenyl phosphate (Dia-APP-TPP)	Dia-APP-TPP-30	30	43	344.9	118.4	3.63	27.8	V-0	40.4	20.5	
	unsaturated polyester	0	66	516.7	123.7		19.8				[27]
Carbon microspheres (CMS)	CMS-3	3	79	459.5	131.2	1.27	21.4				
Phosphorylated chitosan-coated carbon microspheres (PCH@CMS)}	PCH@CMS-3	3	53	419	126.9	0.97	21.7				
	unsaturated polyester	0	25	918.8	91.3		21	NR	70.8	28.5	[97]
Dimelamine pyrophosphate (DMPY)/aluminium diethylphosphinate (ADP)	DMPY-15	15	13	406.6	67.9	1.58	25.6	NR	52	18.3	
Dimelamine pyrophosphate (DMPY)/aluminium diethylphosphinate (ADP)	DMPY-7.5_ADP-7.5	15	23	401.6	54	3.56	27.9	V-0	55.5	24.2	
	unsaturated polyester	0					21	NR	73.3	58	[98]
Aluminium dialkylphosphinate (AlPi)	Alpi-5	5					25	NR	49.1	36.4	
Aluminium dialkylphosphinate (AlPi)	Alpi-15	15					27.5	V-2	45.2	26.2	
Aluminum dialkylphosphinate (AlPi)	Alpi-25	25					29.5	V-0	35.5	20.1	
	unsaturated polyester	0					19.8	NR			[99]
Dimethyl methylphosphonate (DMMP/Aluminium hydroxide (ATH)	DMMP-8.6_ATH-4.3	12.9					24.8	V-2			
Dimethyl methylphosphonate (DMMP/Aluminium hydroxide (ATH)	DMMP-8.3_ATH-8.3	16.6					24.9	V-2			
Dimethyl methylphosphonate (DMMP/Aluminium hydroxide (ATH)	DMMP-8_ATH-12	20					25.5	V-1			
Dimethyl methylphosphonate (DMMP/Aluminium hydroxide (ATH)	DMMP-7.8_ATH-15.3	23.1					25.6	V-1			
Dimethyl methylphosphonate (DMMP/Aluminium hydroxide (ATH)	DMMP-7.4_ATH-18.5	25.9					25.8	V-1			
Dimethyl methylphosphonate (DMMP/Aluminium hydroxide (ATH)/Ammonium polyphosphate (APP)	DMMP-7.8_ATH-11.7_APP-2.3	21.8					26.7	V-1			
Ammonium polyphosphate (APP)/Aluminium hydroxide (ATH)/Dimethyl methylphosphonate (DMMP)	APP-7.6_ATH-11.4_DMMP-4.5	23.5					27.8	V-1			
Ammonium polyphosphate (APP)/Aluminium hydroxide (ATH)/Dimethyl methylphosphonate (DMMP)	APP-7.4_ATH-11.2_DMMP-6.7	25.3					29.8	V-0			
Ammonium polyphosphate (APP)/Aluminium hydroxide (ATH)/Dimethyl methylphosphonate (DMMP)	APP-7.3_ATH-10.9_DMMP-8.7	26.9					30.1	V-0			
Ammonium polyphosphate (APP)/Aluminium hydroxide (ATH)/Dimethyl methylphosphonate (DMMP)	APP-7.1_ATH-10.7_DMMP-10.7	28.5					27.5	V-0			
	unsaturated polyester	0					18.9	NR	50.9	12.6	[100]
Dimethylmethylphosphonate (DMMP)/organic modified Ammonium polyphosphate (APP)/Montmorillonite (MMT)	DMMP-10_ APP-17_MMT-3	30					30.5	V-1	36	16.3	
Dimethylmethylphosphonate (DMMP)/organic modified Ammonium polyphosphate (APP)/Montmorillonite (MMT)/Zinc borate (ZB)	DMMP-10_ APP-17_MMT-1_ZB-2	30					31.3	V-0	44.2	21.2	
Dimethylmethylphosphonate (DMMP)/organic modified Ammonium polyphosphate (APP)/Montmorillonite (MMT)/Zinc borate (ZB)	DMMP-10_ APP-15_MMT-3_ZB-2	30					30.5	V-0	42.2	20.1	
	unsaturated polyester		24	825	131			NR			[101]
Aluminium Trihydrate (ATH)	ATH-40	40	55.3	336.8	121.7	6.075		NR			
Aluminium Trihydrate (ATH)/Aluminium hypophosphite (AHP)	ATH-37_AHP-8	45	58	254.4	92.2	11.14		V-0			
Aluminium Trihydrate (ATH)/Zinc diethylphosphinate (DEPZn)	ATH-37_DEPZn-8	45	56.5	265.3	95.1	10.08		V-0			
	unsaturated polyester	0	44	750.8	128.4		21	NR			[102]
Piperazine pyrophosphate (PPAP)	PPAP-16	16	32	380.9	88.59	2.078	26.6	V-1			
Piperazine pyrophosphate (PPAP)	PPAP-18	18	25	293.3	73.83	2.529	29.8	V0			
Piperazine pyrophosphate (PPAP)	PPAP-20	20					31.1	V-0			
Piperazine pyrophosphate (PPAP)	PPAP-22	22					31.4	V-0			
	unsaturated polyester	0	93	501.4	131.6		21			22.9	[103]
Ammonium polyphosphate Montmorillonite nano compound (AM)	AM-15	15	97	217	51.5	6.158	26.7			19.7	
Ammonium polyphosphate Montmorillonite nano compound (AM)/boron silicate-based graphene oxide (B-Si@GO)	AM-14.95_B-Si@GO-0.05	15					27.2				
Ammonium polyphosphate Montmorillonite nano compound (AM)/boron silicate-based graphene oxide (B-Si@GO)	AM-14.9_B-Si@GO-0.1	15	99	138	31	16.42	28.5			24	
Ammonium polyphosphate Montmorillonite nano compound (AM)/boron silicate-based graphene oxide (B-Si@GO)	AM-14.85_B-Si@GO-0.15	15					28.2				
	unsaturated polyester	0	25	704.8	127.3		19.9	NR	74	41.2	[104]
Ammonium polyphosphate (APP)	APP-30	30	39	260	65.1	8.269	36.3	V-0	34.2	19.4	
Ammonium polyphosphate (APP)/ferric oxide (Fe_2_O_3_)	APP29.5_Fe_2_O_3_-0.5	30	34	259.1	64.7	7.279	39.2	V-0	41.8	21.3	
Ammonium polyphosphate (APP)/Antimony oxide (Sb_2_O_3_)	APP29.5_Sb_2_O_3_-0.6	30	34	295.2	69	5.99	39.4	V-0	51.4	23.8	
Ammonium polyphosphate (APP)/Aluminium oxide (Al_2_O_3_)	APP29.5_Al_2_O_3_-0.7	30	34	261.8	64.3	7.248	40.6	V-0	51.2	22.2	
	unsaturated polyester	0	51	743.19	100.36					37.1	[105]
Novel graphene like nanometal MAX (Ti_3_AlC_2_)	MAX-2	2	48.2	661.11	97.59	1.093				38.7	
Novel graphene like nanometal MXENE(Ti_3_C_2_T_x_)	MXENE-2	2	37.4	523.4	85.5	1.222				41.6	
	unsaturated polyester		62	520	139.8		19	NR	64.9	42.5	[106]
Aluminium hypophosphite (AHP)	AHP-29.5	29.5	73	224.5	79.9	4.772	24.3	V-1	40.7	24.6	
Chlorinated paraffin (CP)	CP-29.5	29.5	39	324	108.9	1.296	23.6	V-2	15.6	7.9	
Chlorinated paraffin (CP)/Aluminium hypophosphite (AHP)	CP_AHP 1:2	29.5	53	278.3	85.8	2.602	27.5	V-0	27.4	16.7	
Aluminium hypophosphite coated with chlorinated paraffin (CP@AHP 1:2)	CP@AHP-29.5	29.5	47	216.5	75.3	3.38	28.5	V-0	51.4	32.1	
	unsaturated polyester		39	562.8	143.2		19.8	NR	79.9	29.3	[107]
Intumescent Flame Retardant IFR (APP-pentaerythritol PER-Melamine MEL (3:1:1)	IFR (APP-MEL-PER)-24.5	24.5	22	263.9	97.7	1.466	27	V-0	32.6	17.7	
Intumescent Flame Retardant IFR (silane treated APP-pentaerythritol PER-Melamine MEL (3:1:1)	IFR (STAPP-MEL-PER)-24.5	24.5	20	258.3	120.7	1.186	41.5	V-0	35.8	20.7	

**Table 6 materials-14-01181-t006:** Cone colorimetry data (TTI, PHRR, THR), Calculated FRI value, LOI, UL-94, FS and TS for Fabric composites filled with a wide variety of FRs. The designation in the second column refers to the type of matrix followed by fabric type and its wt.% content finally FR type and its wt.% content for example M_F-30_FR_1_-2_FR_2_-5 this means matrix M reinforced with 30 wt.% of Fabric F and incorporated by 2 wt.% of flame retardant type FR_1_ and 5 wt.% of flame retardant type FR_2_.

Matrix	FR	Designation	FR wt.%	TTI (s)	PHRR (KW/m2)	THR (MJ/m^2^)	FRI	LOI	UL-94	FS (MPa)	TS (MPa)	Ref.
Epoxy_Glass Fabric	Nano clay modified by an organic surfactant (35–40 wt.%)	Epoxy_GF	0	44	818	28.83	1.0				366	[108]
	Nano clay modified by an organic surfactant (35–40 wt.%)	Epoxy_GF_NC-1	1	31.1	558	26.42	1.1				387	
	Nano clay modified by an organic surfactant (35–40 wt.%)	Epoxy_GF_NC-3	3	32.1	570	25.48	1.2				408	
	Nano clay modified by an organic surfactant (35–40 wt.%)	Epoxy_GF_NC-5	5	33.5	533	24.83	1.4				405	
Epoxy resin	Epoxy	Epoxy	0	23	1910	61	1.0		NR	102	44.5	[109]
	Melamine coated ammonium polyphosphate APP	M-APP-29.7	29.7	24	281	23	18.8		V-0	67	34.1	
	Melamine coated ammonium polyphosphate a APP/Talc	M-APP-19.9_Talc-9.73	29.7	28	357	24	16.6		V-0	62.7	31.1	
Epoxy_Glass Fabric	Melamine coated ammonium polyphosphate APP	Epoxy_GF-53.8	0	24	451	37	1.0		NR	400	339	
	Melamine coated ammonium polyphosphate APP	Epoxy_GF-50.8_M-APP-14.6	14.6	22	233	11	6.0		V-1	386	324	
	Melamine coated ammonium polyphosphate APP/Talc	Epoxy_GF-50.3_M-APP-9.93_Talc-4.84	4.84	21	169	16	5.4		NR	425	280	
Epoxy resin	Epoxy	Epoxy	0	53	1076	91	1.0	22	NR			[110]
	N, N’-diamyl-pphenylphosphonicdiamide(P-MA)	P-MA-5	5	50	469	75	2.6	32	V-1			
	N, N’-diamyl-pphenylphosphonicdiamide(P-MA)	P-MA-8	8	45	405	71	2.9	33	V-1			
	N, N’-diamyl-pphenylphosphonicdiamide(P-MA)	P-MA-12	12	39	363	68	2.9	36	V-0			
	N, N’-diamyl-pphenylphosphonicdiamide(P-MA)	Epoxy_GF-30	0	65	864	56	1.0	25	NR			
	N, N’-diamyl-pphenylphosphonicdiamide(P-MA)	Epoxy_GF-30_P-MA-9.5	9.5	40	400	41	1.8	33	V-0			
	Melanine coated ammonium polyphosphate	Epoxy_GF-54.6	0	34	421.2	37.8	1.6			417		[111]
	Melanine coated ammonium polyphosphate	Epoxy_GF-54.6_APP-9	9	20	269.2	23.7	2.3			411		
Epoxy_Carbon fabric	Graphene grafted with 9,10-dihydro-9-oxa-10-phosphaphenantrene-10-oxide(G-DOPO)	Epoxy_CF-70	0	91	383	93	1.0	18.5	NR			[112]
	Graphene grafted with 9,10-dihydro-9-oxa-10-phosphaphenantrene-10-oxide(G-DOPO)	Epoxy_CF-70_G-DOPO-0.5	0.5	90	311	87	1.3	27.7	V-1			
	Graphene grafted with 9,10-dihydro-9-oxa-10-phosphaphenantrene-10-oxide(G-DOPO)	Epoxy_CF-70_G-DOPO-1	1	17	274	79	0.3	28.2	V-1			
	Graphene grafted with 9,10-dihydro-9-oxa-10-phosphaphenantrene-10-oxide(G-DOPO)	Epoxy_CF-70_G-DOPO-3	3	15	234	70	0.4	28	V-1			
Epoxy resin	Epoxy	Epoxy		38	943	60.3	1.0					[113]
	Layered double hydroxide LDH (anionic unmodified clay)	RS-LD-NC-5	5	35	578	58.4	1.6					
	Layered double hydroxide LDH organic treated (anionic modified clay)	RS-FR-NC-5	5	38	453	66.5	1.9					
	Cationic montmorillonite (cationic unmodified clay)	RS-N2-NC-5	5	33	823	61.7	1.0					
	Cationic montmorillonite) (cationic modified clay)	R5-N116-NC-5	5	38	717	58.6	1.4					
	Carbon nanotube (CNT)	CNT-1	1	26	673	53.8	1.1					
	Chemical treated carbon nanotube with with carboxylic acid functionalization	CT-CNT-1	1	32	837	57.4	1.0					
	Thermally oxidized carbon nanotube (T-CNT)	T-CNT-1	1	25	585	56.6	1.1					
	Aluminium trihydroxide (ATH)	ATH-5	5	35	617	59.2	1.4					
	Ammonium polyphosphate (APP)	APP_5	5	36	543	58.8	1.7					
Epoxy_Carbon fabric	Carbon fibre reinforced epoxy	Epoxy_CF-54.8	0	28	349	20.4	1.0					
	Anionic unmodified clay	Epoxy_CF-54.7_RS_RS-LD-NC-5	5	22	343	21.9	0.7					
	Anionic modified clay	Epoxy_CF-57.7_RS-FR-NC-5	5	21	310	23	0.7					
	Carbon nanotube	Epoxy_CF-56.7_CNT-1	1	27	396	22.7	0.8					
	Chemical treated carbon nanotube	Epoxy_CF-55.2_CT-CNT-1	1	26	411	21.7	0.7					
	Thermal treated carbon nanotube	Epoxy_CF-58.3_T-CNT-1	1	27	471	22.2	0.7					
	Alumina trihydroxide (ATH)	Epoxy_CF-55.5_ATH-5	5	22	417	22.6	0.6					
	Ammonium polyphosphate (APP)	Epoxy_CF-54.7_APP-5	5	24	345	18.6	1.0					
Epoxy_Hemp Fabric	Ammonium Polyphosphate (APP)	Epoxy_Hemp-35	0	21.2	720.5	68	1.0			128.3		[114]
	Ammonium Polyphosphate (APP)	Epoxy_Hemp-35_APP-3.15	3.15	20.3	375.3	42	3.0			127.1		
	Ammonium Polyphosphate (APP)	Epoxy_Hemp-35_APP-8.88	8.88	18.1	293.8	33	4.3			131.3		
	Ammonium Polyphosphate (APP)	Epoxy_Hemp-35_APP-16.32	16.32	21	186.7	27	9.6			127.3		
Epoxy_Flax	melamine coated ammonium polyphosphate	Epoxy_Flax-37.9	0	16	619.6	68.5	1.0			124		[111]
	melamine coated ammonium polyphosphate	Epoxy_Flax-37.9_APP-7.58	7.58	25	269.4	40.2	6.1			116		
Unsaturated Polyester_Glass fabric	polyester-Glass fabric	Polyester_GF-50	0	134	339.77	92.4	1.0	21	NR			[115]
	Alumina trihydrate (ATH)	Polyester_GF-50_ATH-18.5	18.5	158	278.89	122.5	1.1	23	NR			
	Decabromodiphenyl ether (DBDE)	Polyester_GF-50_DBDE-3.25	3.25	126	282.68	105.9	1.0	22	NR			
	Alumina trihydrate (ATH)/decabromodiphenyl ether (DBDE)	Polyester_GF-50_ATH-18.5_DBDE-3.25	21.75	147	277.88	115.5	1.1	25	V-0			
	Decabromodiphenyl ether (DBDE)/antimony trioxide (Sb_2_O_3_)	Polyester_GF-50_DBDE-6.5_Sb_2_O_3_-2.15	8.65	165	214.03	53.8	3.4	33	V-0			
	DBDE)/antimony trioxide (Sb_2_O_3_)	Polyester_GF-50_DBDE-6.5_Sb2O3-3.25	13	200	155.82	24.7	12.2	36	V-0			
	Alumina trihydrate (ATH)/decabromodiphenyl ether (DBDE)/antimony trioxide (Sb_2_O_3_)	Polyester_GF-50_ATH-18.5_DBDE-6.5_Sb2O-3-2.15	27.15	220	154.27	40.5	8.2	33	V-0			
	Alumina trihydrate (ATH)/decabromodiphenyl ether (DBDE)/antimony trioxide (Sb_2_O_3_)	Polyester_GF-50_ATH-18.5_DBDE-9.75_Sb2O3-3.25	31.5	181	140.62	36.4	8.3	37	V-0			
	Alumina trihydrate (ATH)	Polyester_GF-50_ATH-23.8	23.8	134	339.77	92.4	1.0	26	V-0			
	Alumina trihydrate (ATH)	Polyester_GF-50_ATH-29.25	29.25	158	278.89	122.5	1.1	29	V-0			
Epoxy/Unsaturated polyester blend	Epoxy/Unsaturated polyester blend	Epoxy-95_UP-5	1	61	829.2	141.7		20	V-2	119	65	[116]
	Nanoclay, bis(2-hydroxy-ethyl) methyl tallow ammonium	Epoxy-95_UP-5_ NC-1	1	66	647.2	119.5	1.66	24	V-1	131	74	
Epoxy/UP_Sisal Fabric	Epoxy/UP_alkali-silane treated sisal fibre	Epoxy-95_UP-5_ASTF-30	0	65	610.9	110.8		25	V-1	180	119	
	Nanoclay, bis(2-hydroxy-ethyl) methyl tallow ammonium	Epoxy-95_UP-5_ NC-1_ASTF-30	1	64	583.3	104.3	1.09	27	V-1	191	128	
Epoxy/Novolac type cyanate ester (CE) blend	Cyanate ester	CE	0	26	156	15.5		30	HB			[117,118]
	Epoxy	Epoxy	0	40	743	91		23	HB			
	Epoxy-Novolac type cyanate ester	Epoxy-80_CE-20	0	50	471	59.5	3.02	33	HB			
	Epoxy-Novolac type cyanate ester	Epoxy-60_CE-40	0	50	238	55.1	6.44	28	HB			
	Epoxy-DOPO	Epoxy-DOPO-13.94	13.94	32	477	65.1	1.74	29	V-1			
	Epoxy-Novolac type cyanate ester_DOPO	Epoxy-80_CE-20_DOPO-13.94	13.94	42	261	49	5.55	42	V-0			
	Epoxy-Novolac type cyanate ester_DOPO	Epoxy-70_CE-30_DOPO-13.94	13.94	50	207	42	9.72	40	V-0			
	Epoxy-Novolac type cyanate ester_DOPO	Epoxy-60_CE-40_DOPO-13.95	13.94	53	195	36.3	12.7	43	V-0			
	Epoxy-Novolac type cyanate ester_DOPO	Epoxy-80_CE-20_DOPO-20.9	20.9	27	218	50.3	4.16	40	V-0			
	Epoxy-Novolac type cyanate ester_DOPO	Epoxy-75_CE-25_DOPO-20.9	20.9	45	218	46	7.59	42	V-0			
	Epoxy-Novolac type cyanate ester_DOPO	Epoxy-60_CE-40_DOPO-20.9	20.9	44	234	47.5	6.69	45	V-0			
Epoxy/Novolac type cyanate este _Carbon fabric	Novolac type cyanate ester (Primaset PT-30)/carbon fabric	CE_CF-55	0	80	84	9.8		58	V-0			
	Epoxy/Carbon fabric	Epoxy_CF-56	0	55	176	37.9		33	HB	1203	912.6	
	Epoxy-Novolac type cyanate ester (CE)/carbon fabric (CF)	Epoxy-80_CE-20_CF-55	0	51	162	29.9	1.28	41	HB	1240	1040	
	Epoxy-Novolac type cyanate ester (CE)/carbon fabric (CF)	Epoxy-60_CE-40_CF-55	0	87	134	21.8	3.61	42	V-0	1238	844.1	
	Epoxy-Novolac type cyanate ester (CE)/carbon fabric (CF)-DOPO 2% P	Epoxy-60_CE-40_CF-55_DOPO(2%p)-6.27	6.273	72	101	20.1	4.3	46	V-0	1056	861.2	
	Epoxy-Novolac type cyanate ester (CE)/Carbon fabric (CF)-DOPO 3% P	Epoxy-60_CE-40_CF-55_DOPO(3%p)-9.4	9.4	70	84	18.7	5.4	48	V-0	1149	715.2	
saturated polyester/phenolic resin blend	unsaturated polyester (UP)	UP	0	40	1053	78.9						[119,120]
	UP/Solvent based phenolic (PH–S)	Up-70_PH-S-30	0	31	630	62.3	1.64					
	UP/Solvent based phenolic (PH–S)	Up-50_PH-S-50	0	31	568	48.4	2.34					
	UP/Epoxy functionalised phenolic (PH-Ep)	Up-70_PH-Ep-30	0	39	885	54.3	1.69					
	UP/Epoxy functionalised phenolic (PH-Ep)	Up-50_PH-Ep-50	0	34	682	49.6	2.09					
	UP/Allyl functionalised phenolic (PH–Al)	Up-70_PH-Al30	0	54	955	70.7	1.66					
	UP/Allyl functionalised phenolic (PH–Al)	Up-50_PH-Al-50	0	57	828	61	2.34					
Unsaturated polyester/phenolic blend _Glass fabric	glass fabric/unsaturated polyester (59% wt.)	Up_GF-59	0	38	479	30.3					375	
	glass fabric/UP/Solvent based phenolic (PH–S)	Up:PH-S-70:30_GF-59	0	39	418	26.2	1.36				281	
	glass fabric/UP/Solvent based phenolic (PH–S)	Up:PH-S-50:50_GF-55	0	34	365	23.5	1.51				256	
	glass fabric/UP/Epoxy functionalised phenolic (PH-Ep)	Up:PH-Ep-70:30_GF-59	0	42	461	25.3	1.38				298	
	glass fabric/UP/Epoxy functionalised phenolic (PH-Ep)	Up:PH-Ep-50:50_GF-67	0	32	448	19.2	1.42				268	
	glass fabric/UP/Allyl functionalised phenolic (PH–Al)	Up-PH-A-70:30_GF-59	0	46	443	28.7	1.38				317	
	glass fabric/UP/Allyl functionalised phenolic (PH–Al)	Up:PH-Al-50:50_GF-58	0	48	415	25.8	1.71					
Epoxy_ Hemp Fabric	Hemp fabrics/epoxy	Epoxy_Hemp-25	0	55	754	61.3				109		[121]
	Waterglass treated Hemp fabrics/epoxy composite	Epoxy_WGT-Hemp-25	0	39	642	64.2	0.8			92		
	Hemp fabrics/epoxy/Ammonium polyphosphate (APP)	Epoxy_Hemp-25_APP-15	15	46	259	34.4	4.34			110		
	Hemp fabrics treated with waterglass/epoxy/Ammonium polyphosphate (APP)	Epoxy_WGT-Hemp-26_APP-15	15	44	232	40.1	3.97			94		
Epoxy _Carbon fabric	Carbon fibre reinforced epoxy	Epoxy_CF-46	47.5	54	508.3	47.8				977.2		[122]
	Carbon Fibre decorated by d by bio-based polyelectrolyte complexes (PEC) of chitosan and ammonium polyphosphate	Epoxy_CF-46-PEC-5.2	5.2	51	358	44	1.46			916		
	Carbon Fibre decorated by d by bio-based polyelectrolyte complexes (PEC) consisting of chitosan and ammonium polyphosphate	Epoxy_CF-46-PEC-6.9	6.9	50	307.5	39.6	1.85			907		
	Carbon Fibre decorated by d by bio-based polyelectrolyte complexes (PEC) consisting of chitosan and ammonium polyphosphate	Epoxy_CF-46-PEC-8.1	8.1	49	255.9	35.5	2.43			863.6		
Epoxy_Glass fabric	Epoxy-Glass fabric	Epoxy_GF	0					22.5	NR			[123]
	Organo montmorillonite clay (OMMT)	Epoxy_GF_OMMT-2	2					22.7	NR			
	Organo montmorillonite clay (OMMT)-brominated flame retardants decabromodiphenyl oxide (DBDPO)	Epoxy_GF_OMMT-2_DBDPO-10	12					23.9	NR			
	Organo montmorillonite clay (OMMT)-brominated flame retardants decabromodiphenyl oxide (DBDPO)	Epoxy_GF_OMMT-2_DBDPO-20	22					27.4	NR			
	Organo montmorillonite clay (OMMT)-brominated flame retardants decabromodiphenyl oxide (DBDPO)	Epoxy_GF_OMMT-2_DBDPO-30	32					32	NR			
	Organo montmorillonite clay (OMMT)-brominated flame retardants decabromodiphenyl oxide (DBDPO)	Epoxy_GF_OMMT-2_DBDPO-40	42					38	V-1			
	Organo montmorillonite clay (OMMT)-brominated flame retardants decabromodiphenyl oxide (DBDPO)	Epoxy_GF_OMMT-2_DBDPO-50	52					39.9	V-0			
Epoxy_Glass fabric	Epoxy-Glass fabric	Epoxy_GF-57.3	0					18.5	NR		197.2	[124]
	Conventional addition of melamine polyphosphate (MPP)	Epoxy_GF-57.3_MPP-7.1	7.1					31.2	V-0		157.4	
	In situ dispersed melamine polyphosphate (Insitu-MPP)	Epoxy_GF-57.3_Insitu-MPP-7.1	7.1					34.3	V-0		178.3	
Epoxy_Flax fabric	2-layer flax fabric-reinforced Epoxy	Epoxy-2 Layer Flax	0					21.3	NR		5.42	[125]
	4-layer flax fabric-reinforced Epoxy	Epoxy-4 Layer Flax	0					23.3	NR			
	2-layer flax fabric-reinforced Epoxy/Ammonium polyphosphate	Epoxy-2 Layer Flax_APP-10	10					22.4	NR		4.13	
	2-layer flax fabric-reinforced Epoxy/Ammonium polyphosphate	Epoxy-2 Layer Flax_APP-20	20					25.5	V-1		4.53	
	2-layer flax fabric-reinforced Epoxy/Ammonium polyphosphate	Epoxy-2 Layer Flax_APP-30	30					30.3	V-0			
	2-layer flax fabric-reinforced Epoxy/aluminium hydroxide (ALH)	Epoxy-2 Layer Flax_ALH-20	20					22.5	NR		4.29	
	3-layer flax fabric-reinforced Epoxy/aluminium hydroxide (ALH)	Epoxy-2 Layer Flax_ALH-30	30					23.5	NR		4.57	
	4-layer flax fabric-reinforced Epoxy/aluminium hydroxide (ALH)	Epoxy-2 Layer Flax_ALH-40	40					24.5	NR		4.46	
Unsaturated Polyester_Glass fabric	Polyester-Glass fabric	Polyester-GF									75.5	[126]
	Aluminium trihydroxide (ATH)	Polyester_GF_ATH-40%	40					33	V0		73.2	
	Aluminium trihydroxide (ATH)/Expandable Graphite (EG)	Polyester_GF_ATH-36%_EG-4%	40					34	NR			
	Aluminium trihydroxide (ATH)/Expandable Graphite (EG)	Polyester_GF_ATH-32%_EG-8%	40					35	NR			
	Aluminium trihydroxide (ATH)/Expandable Graphite (EG)	Polyester_GF_ATH-20%_EG-20%	40					36	V0		65.1	
	Aluminium trihydroxide (ATH)/Ammonium Polyphosphate (APP)	Polyester_GF_ATH-36%_APP-4%	40					35	NR		72.1	
	Aluminium trihydroxide (ATH)/Ammonium Polyphosphate (APP)	Polyester_GF_ATH-20%_APP-20%	40					43	V-0		73.1	
	Aluminium trihydroxide (ATH)/Ammonium Polyphosphate (APP)/Expandable Graphite (EG)	Polyester_GF_ATH-28%_APP-4%_EG-8%	40					37	NR		70.6	
	Aluminium trihydroxide (ATH)/Ammonium Polyphosphate (APP)/Expandable Graphite (EG)	Polyester_GF_ATH-28%_APP-8%_EG-4%	40					37	NR		69	
	Aluminium trihydroxide (ATH)/Ammonium Polyphosphate (APP)/Expandable Graphite (EG)	Polyester_GF_ATH-20%_APP-12%_EG-8%	40					43	V-0		72.1	
Epoxy_Glass Fabric	Epoxy/glass fabric	Epoxy_GF-50	0					25.9			351	[127]
	Ammonium polyphosphate (APP)	Epoxy_GF-50_APP-5	5					29.1			365	
	Ammonium polyphosphate (APP)	Epoxy_GF-50_APP-10	10					29.7			352	
	Ammonium polyphosphate (APP)	Epoxy_GF-50_APP-20	20					30.1			358	
	Melamine polyphosphate (PNA)	Epoxy_GF-50_PNA-5	5					32.5			388	
	Melamine polyphosphate (PNA)	Epoxy_GF-50_PNA-10	10					31.3			365	
	Melamine polyphosphate (PNA)	Epoxy_GF-50_PNA-20	20					31.5			343	
Epoxy _Glass fabric	Epoxy/glass fabric	Epoxy_GF-50						21	NR	310.8	184	[128]
	Reactive 9,10-Dihydro-9-oxa-10-phosphaphenanthrene-10-oxide (DOPO)	Epoxy_GF-DOPO-4.2	4.2					24	NR	241.5	142	
Unsaturated Polyester_Glass fabric	Glass fabric Grafted with silane treated DOPO	Epoxy_GGF	4.2					31	V-1	312.5	187	[129]
	Unsaturated polyester/glass fabric	UP_GF-20		85	439	85.2		21.7		128		
	Na-Montmorillonite nanoclay	UP_GF-20_Na-MMT-3	3	79	412	82.5		22.2		155		
	Silane treated Na-Montmorillonite nanoclay	UP_GF-20_Silane Na-MMT-2.54	2.54	79	409	84.5		22.1		162		
	Montmorillonite nanoclay	UP_GF-20_MMT-2.3	2.3	87	434	84.7		21.6		147		
	Silane treated Montmorillonite nanoclay	UP_GF-20_silane MMT-3	3	82	373	85		21.5		166		
	Calcium carbonate	UP_GF-20_CaCO_3_	3	86	446	88.3		21		143		
Unsaturated Polyester_Glass fabric	Unsaturated polyester/glass fabric	UP_GF-50						21		238.8		[130]
	Antimony oxide (AO)	UP_GF-35_AO-15	15					25		230.8		
	Antimony oxide (AO)/Fly ash (FA)	UP_GF-35_AO-10_FA-5	15					27		238.7		
	Antimony oxide (AO)/Fly ash (FA)	UP_GF-35_AO-5_FA-10	15					29		230		
	Fly ash (FA)	UP_GF-35_FA-15	15					31		225		
	Antimony oxide/Hydroxyapatite (HA)	UP_GF-35_AO-10_HA-5	15					28		240		
	Antimony oxide/Hydroxyapatite (HA)	UP_GF-35_AO-5_HA-10	15					30		245		
	Hydroxyapatite (HA)	UP_GF-35_HA-15	15					32		247		
	Antimony oxide/Zinc borate (ZB)	UP_GF-35_AO-10_ZB-5	15					32		242		
	Antimony oxide/Zinc borate (ZB)	UP_GF-35_AO-5_ZB-10	15					34		246		
	Zinc borate (ZB)	UP_GF-35_ZB-15	15					37		247		

## Data Availability

The data that support the findings of this study are available from the corresponding author, [A.E.], upon reasonable request.

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
