# Peer review of "Towards Selection Charts for Epoxy Resin, Unsaturated Polyester Resin and Their Fibre-Fabric Composites with Flame Retardants"

_materials, 2021, doi:10.3390/ma14051181_

Round 1

Reviewer 1 Report

The manuscript reviews and compares the flame retardant performances of different systems used for epoxy and unsaturated resins.

I think the manuscript could be useful for researchers in the field, but it seems that the authors didn’t pay much attention when they wrote the review. Being a long document, filled with data and abbreviations is difficult to read unless everything is very clear and easy to follow. So I recommend the manuscript for publication after minor but careful review.

I list below some observations:

  1. I recommend not using the term “epoxy” without adding resin, or polymer, or material, or composites, or something similar.
  2. Please check all the abbreviations in the text. For example sometimes is UL-94, other times Ul-94. Other abbreviations appear without any explanation of what they mean. See FRP (appears for the first time in r. 40), PP, PMMA, EVA, PLA (r. 162), etc.
  3. r. 38: “…steel and concert…”????
  4. r. 51 and 54: Please reformulate these sentences so they could be more clear: “However, the benefits of using polymeric materials, but in return more combustible materials are added and more fire risks that threat human life and loss of property take place [18, 19].” and “In some studies, the several tons of polymers used in overhead bin, internal panels, seat fabric and cushions in aircraft’s passenger compartment act as a fuel source for fire results from airplane crashes and this results in reducing the time of escape [9, 20].”
  5. r. 61: “…to reduce smoke and to delay the time of flashover, subsequently providing sufficient time for people to escape…”
  6. r. 148: “FRs can also be classified either as additive when FR compounds are directly incorporated to polymer matrix or as reactive FR when FR functional groups are part of the molecular structure of polymers” or else that is more clear.
  7. r. 257: “…there are three more points in the Excellent…”
  8. r. 312: …”Figures 4d,4d and 4f” ????
  9. r. 70 (section 2): Please reformulate:  “Although the calculated Flame retardancy index can be used as a reliable measure in comparing the 71 performance of different FR/polymer system, but FRI value is based on the data collected from cone calorimetry 72 and from the cost wise perspective conducting Ul-94 and LOI tests are not as expensive as cone calorimetry test.”
  10. r. 145: properties
  11. r. 152: …”as a physical…”
  12. r. 163: …”that are located…”
  13. r. 201: …”combination of different types….??????
  14. r. 210: …”both….and….”

Reviewer 2 Report

this review provided detail information about the Flame Retardants resin and composites, the information is very useful, it could be published after revision.

  1. the title is not easy to understand, should be modified.
  2. the introduction part is too long, some of the information should be listed seperated
  3. too much data in figure 3, make it not easy to read, also in Figure 4-7
  4. the English need improvement
  5. the references should keep the same format

Reviewer 3 Report

The authors present an exhaustive review about the use of flame retardants (FR) in thermosets and fiber-reinforced thermosets, with evident interest to the readers. They show is a noticeable and useful bibliographic data collection and there is also an effort to present data in a useful way, although it has not been completely achieved a clear presentation of the factors that influence the performance of the different FR in these materials. For example, the different physico-chemical nature of the graphene used in different works (obtention and reduction method employed configure the chemical structure and degree of reduction) is obviated, which is usually the reason for contradictory results in the use of this nanofiller, and similar issues may be present with another FR additives. In the same way, there is not attention done to the method of preparation of the thermosets or composites, or how the FR have been dispersed on the material, which may have influence on the final properties. A clearer description of the different indices used to assess flame retardancy would also be appreciated. Those may be considered as suggestions to increase the utility of the review.
There are also two more formal factors that must be solved before publication: i) The bibliography must be thoroughly reviewed. In a non-exhaustive check, I have found that more than 10% of the bibliographic references are incomplete. For example, in ref. 52 "Polymer Degratation and Stability", is missing  and similarly in ref. 56 ("Annual Transactions Of The Nordic Rheology Society"). Some references are directly wrong, for example ref. 39 (Polymers 2019, 11 (3), 407). 
ii) Although the quality of the English used was not analyzed in depth, there are numerous gross errors in the text that must be refined. For example "concert", "retradancy" or "preformce". There are also errors in some verb forms and in the use of the plural. 
The authors should check the entire manuscript.

Reviewer 4 Report

The manuscript entitled, “towards Selection Charts for Thermosets and their Fibre-Fabric Composites with Flame Retardants”, is in the right scope of the journal and a much-needed review of current increased flame retardant composites. My comments are below;

  1. What are the limitations of certain flame retardants which are still market-available, can you comment on that?
  2. The incorporation mechanism of flame retardants should be discussed, especially in textiles.
  3. Perhaps some sentences on organophosphorus with their advantages, add this article for such flame retardants. (1) Process optimization of eco-friendly flame retardant finish for cotton fabric: A response surface methodology approach and (2) Optimizing Organophosphorus Fire Resistant Finish for Cotton Fabric Using Box-Behnken Design
  4. Maybe some environmental aspect should be discussed with such papers, a) Life cycle assessment of flame retardant cotton textiles with optimized end-of-life phase.
  5. The conclusion is written well, future perspectives are too general, it should be pinpointed with details and all technicalities.

Round 2

Reviewer 3 Report

The manuscript has been improved and submitted to intensive english language edition, so it is easy to read now. The bibliography, as expected, has been also exhaustively revised and surprisingly several references have completely changed.
I consider the manuscript may be published in its present form, but recommend a final check it because some mistakes remain (for example, page 3 line 84 Polypropilene appears repeated) 

Reviewer 4 Report

No further comments.